# A Novel MVA-Based HIV Vaccine Candidate (MVA-gp145-GPN) Co-Expressing Clade C Membrane-Bound Trimeric gp145 Env and Gag-Induced Virus-Like Particles (VLPs) Triggered Broad and Multifunctional HIV-1-Specific T Cell and Antibody Responses

**DOI:** 10.3390/v11020160

**Published:** 2019-02-16

**Authors:** Beatriz Perdiguero, Cristina Sánchez-Corzo, Carlos Oscar S. Sorzano, Lidia Saiz, Pilar Mediavilla, Mariano Esteban, Carmen Elena Gómez

**Affiliations:** 1Department of Molecular and Cellular Biology, Centro Nacional de Biotecnología, Consejo Superior de Investigaciones Científicas (CNB-CSIC), Campus de Cantoblanco, 28049 Madrid, Spain; perdigue@cnb.csic.es (B.P.); cscorzo@cnb.csic.es (C.S.-C.); lidiasm1995@gmail.com (L.S.); pilarmediavilla15@hotmail.com (P.M.); 2Biocomputing Unit, Centro Nacional de Biotecnología, Consejo Superior de Investigaciones Científicas (CNB-CSIC), Campus de Cantoblanco, 28049 Madrid, Spain; coss@cnb.csic.es

**Keywords:** HIV-1, MVA vaccine, Env-gp145, Gag-Pol-Nef, VLPs, immunogenicity, CD4 T cells, Tfh, GC B cells, humoral responses

## Abstract

The development of an effective Human Immunodeficiency Virus (HIV) vaccine that is able to stimulate both the humoral and cellular HIV-1-specific immune responses remains a major priority challenge. In this study, we described the generation and preclinical evaluation of single and double modified vaccinia virus Ankara (MVA)-based candidates expressing the HIV-1 clade C membrane-bound gp145(ZM96) trimeric protein and/or the Gag(ZM96)-Pol-Nef(CN54) (GPN) polyprotein that was processed to form Gag-induced virus-like particles (VLPs). In vitro characterization of MVA recombinants revealed the stable integration of HIV-1 genes without affecting its replication capacity. In cells that were infected with Env-expressing viruses, the gp145 protein was inserted into the plasma membrane exposing critical epitopes that were recognized by broadly neutralizing antibodies (bNAbs), whereas Gag-induced VLPs were released from cells that were infected with GPN-expressing viruses. VLP particles as well as purified MVA virions contain Env and Gag visualized by immunoelectron microscopy and western-blot of fractions that were obtained after detergent treatments of purified virus particles. In BALB/c mice, homologous MVA-gp145-GPN prime/boost regimen induced broad and polyfunctional Env- and Gag-specific CD4 T cells and antigen-specific T follicular helper (Tfh) and Germinal Center (GC) B cells, which correlated with robust HIV-1-specific humoral responses. Overall, these results support the consideration of MVA-gp145-GPN vector as a potential vaccine candidate against HIV-1.

## 1. Introduction

The AIDS (Acquired Immunodeficiency Syndrome) pandemic that is caused by Human Immunodeficiency Virus (HIV) represents a global health problem with enormous disease control dimensions. Thanks to the efforts in strengthening HIV/AIDS prevention programs and the increasing numbers of HIV-infected individuals that have access to the highly effective antiretroviral therapy (HAART), the annual number of new HIV infections has declined by 16% since 2010 (http://www.unaids.org). Nonetheless, at the end of 2016, over 36 million people worldwide were living with HIV/AIDS. As there is no functional cure for HIV infection thus far, an effective HIV vaccine remains the best long-term strategy for preventing viral infection and AIDS.

We have previously reported that poxvirus-based viral vectors are potential prophylactic vaccine candidates against HIV infection [1,2,3,4]. The modified vaccinia virus Ankara-B (MVA-B) vaccine simultaneously co-expressing the HIV-1 (clade B) monomeric gp120 envelope (Env) protein as a cell-released product and Gag-Pol-Nef (GPN) antigens as an intracellular polyprotein unable to form VLPs was safe, well tolerated, and elicited moderate and durable HIV-1-specific T cell and antibody responses when it was assayed in healthy volunteers in homologous regimen [1,2]. The same HIV-1 antigens, but from clade C (MVA-C), were also safe and triggered HIV-1 specific immune responses when used in a prophylactic phase I clinical trial in combination with a DNA vector and a gp140 protein component [5]. Moreover, when these clade C HIV-1 antigens were vectored by the DNA and NYVAC strain, induced strong, broad, and polyfunctional T cell responses in humans when combined in the heterologous DNA prime/NYVAC boost regimen [6]. Commonly, in these studies, the induced T cell responses were predominantly directed against Env, whereas GPN-specific responses were lower and less frequent.

In order to increase the expression levels of the encoded HIV-1 antigens and to achieve more balanced immune responses, a new generation of optimized HIV-1 *env* and *gag-pol-nef* genes were designed and then inserted independently into different backbones, such as DNA vectors and attenuated poxvirus strains (NYVAC and ALVAC) [7,8,9,10,11]. The improved antigens belong to the HIV-1 clade C, which is responsible for approximately 50% of all new infections worldwide. The original GPN polyprotein was further refined to allow for the efficient production and release of virus-like particles and to better balance the relative expression of Gag and Pol-Nef antigens and a trimeric soluble gp140 form was used instead of the monomeric gp120 to more closely resemble the native envelope structure. The new generation of recombinant vectors demonstrated an inducement of an enhanced HIV-1-specific immunogenicity profile in mice [11] and non-human primates (NHPs) [8,9,10,12,13] when combined in homologous or heterologous combination.

Since vaccine-induced protective immunity is critically determined by the HIV-1 Env conformation and Gag-specific cellular response, significant efforts are directed towards generating trimeric Env immunogens that assume native structures and Gag-induced VLPs with enhanced immunogenicity. Here, we generated and characterized single and double MVA-based vectors that expressed the HIV-1 clade C gp145(ZM96) Env as a membrane-bound gp145 trimeric protein and/or the improved Gag(ZM96)-Pol-Nef(CN54) (GPN) polyprotein, which is processed in a way that produces a 55 kDa Gag protein that is able to induce the formation of virus-like particles (VLPs) [11]. The immunogenicity of the double MVA-gp145-GPN virus was evaluated in mice in comparison with single recombinants that individually expressed either gp145(ZM96) Env (MVA-gp145) or Gag(ZM96)-Pol-Nef(CN54) (GPN) polyprotein (MVA-GPN). Based on the broad capacity of membrane-bound gp145 to react with bNAbs and on the balanced HIV-1-specific immune responses that are induced by the double recombinant MVA vector (CD4, Tfh, GC B cells, and IgG2a/IgG1 ratio), our findings suggest a potential role of MVA-gp145-GPN as a relevant vaccine against HIV.

## 2. Materials and Methods

### 2.1. Cells and Viruses

Primary chicken embryo fibroblast (CEF) cells (obtained from pathogen-free 11-day-old eggs; MSD, Salamanca, Spain), DF-1 cells (a spontaneously immortalized CEF cell line) and HeLa cells (human epithelial cervix adenocarcinoma cells) were grown in Dulbecco’s modified Eagle’s medium (DMEM) supplemented with 100 U/mL penicillin/100 µg/mL streptomycin (SIGMA, St. Louis, MO, USA), 2 mM l-glutamine (Merck, Kenilworth, NJ, USA), 0.1 mM non-essential amino acids (SIGMA), 0.5 μg/mL amphotericin B (Fungizone; Gibco-Life Technologies, Waltham, MA, USA) and 10% heat-inactivated fetal calf serum (FCS; SIGMA) for CEF and DF-1 cells or 10% newborn calf serum (NCS; SIGMA) for HeLa cells. The cells were maintained in a humidified air 5% CO_2_ atmosphere at 37 °C.

The viruses that were used in this work included: the attenuated wild-type modified vaccinia virus Ankara (MVA-WT) that was obtained from the Ankara strain after 586 serial passages in CEF cells (kindly provided by G. Sutter); the recombinant MVA-gp145(ZM96) expressing a membrane-bound trimeric HIV-1 clade C ZM96 gp145 protein from the viral thymidine kinase (TK) locus (shortly MVA-gp145); the recombinant MVA-Gag(ZM96)-Pol-Nef(CN54) expressing the optimized Gag(ZM96)-Pol-Nef(CN54) polyprotein, which is processed to produce a 55 kDa Gag protein that is able to induce the formation of VLPs from the viral TK locus (shortly MVA-GPN); and, the recombinant MVA-gp145(ZM96)-Gag(ZM96)-Pol-Nef(CN54) expressing gp145(ZM96) from the viral TK locus and Gag(ZM96)-Pol-Nef(CN54) polyprotein from the viral haemagglutinin (HA) locus (shortly MVA-gp145-GPN). In both of the GPN-expressing vectors, the natural ribosomal (−1) frameshift between Gag and Pol was restored to skew Gag:PolNef expression to approximately 10:1, and the N-terminal myristoylation signal was reintroduced to enable the release of GagPolNef virus-like particles from infected cells [9]. Virus infections were performed with 2% FCS or NCS.

### 2.2. Construction of the Plasmid Transfer Vectors

#### 2.2.1. Construction of the Plasmid Transfer Vector pCyA-gp145(ZM96)

The plasmid transfer vector pCyA-gp145(ZM96) (shortly pCyA-gp145), which was used for the insertion of gp145 antigen into the viral TK locus of MVA-WT, was obtained by standard cloning procedures. The codon optimized *env* gen was amplified by PCR from plasmid plZAW1-gp145-ZM96-DeltaC6 (provided by Ralf Wagner, University of Regensburg) with oligonucleotides gp145TM-U1: (5´-GACTCGAGGCCACCATGGGAGTG-3´) (*Xho*I site underlined) and gp145TM-L1: (5´-TAGCGGCCGCTCAGTAGCCCTG-3´) (*Not*I site underlined) (gp145 PCR product: 2178 bp), digested with *Xho*I and *Not*I, and then cloned into plasmid pCyA (7582 bp) [14] that was previously digested with the same restriction enzymes to generate pCyA-gp145 (9654 bp). The resulting plasmid pCyA-gp145 was confirmed by DNA sequence analysis and it directs the insertion of gp145 antigen into the TK locus of MVA-WT.

#### 2.2.2. Construction of the Plasmid Transfer Vector pHA-Gag(ZM96)-Pol-Nef(CN54)

The plasmid transfer vector pHA-Gag(ZM96)-Pol-Nef(CN54) (shortly pHA-GPN), which was used for the insertion of GPN antigen into the viral haemagglutinin (HA) locus of MVA-gp145 recombinant virus, was obtained by standard cloning procedures. The codon optimized GPN gen was amplified by PCR from plasmid plZAW1-Gag(ZM96)-Pol-Nef(CN54) [11] with oligonucleotides GPN-*Not*I (5’-AATTGCGGCCGCTTACTTGGTCCTGTG-3’) (*Not*I site underlined) and GPN-*Kpn*I (5’-AGAGGTACCGCCACCATGGGAGCCAGAG-3’) (*Kpn*I site underlined) (GPN PCR product: 4070 bp), digested with *Not*I and *Kpn*I, and then cloned into plasmid pHA (6639 bp) that was previously digested with the same restriction enzymes to generate pHA-GPN (10,666 bp). The plasmid pHA has been previously described [15] and it comprises the viral sE/L promoter, a multiple-cloning site, HA flanking regions of MVA genome, and the selectable marker genes for ampicillin and β-glucuronidase. The resulting plasmid pHA-GPN was confirmed by DNA sequence analysis and it directs the insertion of the GPN antigen into the HA locus of MVA-gp145-GPN.

### 2.3. Construction of MVA-Based Recombinant Viruses

For the construction of MVA-gp145 and MVA-GPN recombinant viruses, 3 × 10^6^ DF-1 cells were infected with MVA-WT at a multiplicity of infection (MOI) of 0.05 pfu/cell and then transfected one hour later with 6 µg DNA of pCyA-gp145 or plZAW1-Gag(ZM96)-Pol-Nef(CN54) [11], respectively, using Lipofectamine-2000 (Invitrogen, Carlsbad, CA, USA) and following the manufacturer’s instructions. After 72 h post-infection (h.p.i.), the infected cells were collected, lysed by freeze-thaw cycling, sonicated, and then used for the screening of the MVA-based recombinant viruses. MVA viruses containing gp145 or Gag-Pol-Nef genes and transiently co-expressing the β-gal marker gene were isolated by three consecutive rounds of plaque purification steps in monolayers of DF-1 cells that were stained with 5-bromo-4-chloro-3-indolyl β-D-galactopyranoside (X-Gal; 1.2 mg/mL). After the recombinant viruses expressing gp145 or Gag-Pol-Nef and the β-gal marker have been isolated, further propagation of these MVA-based recombinant viruses leads to the self-deletion of β-gal by homologous recombination events between the short TK left arm repeat and the TK left arm that are flanking the marker. Therefore, in the subsequent three rounds, MVA-based recombinant viruses containing gp145 or Gag-Pol-Nef genes and having deleted the β-gal marker gene were selected by plaque purification screening for non-stained viral plaques in DF-1 cells in the presence of 5-bromo-4-chloro-3-indolyl β-D-galactopyranoside (1.2 mg/mL).

For the construction of MVA-gp145-GPN recombinant virus, 3 × 10^6^ DF-1 cells were infected with MVA-gp145 recombinant virus at 0.05 pfu/cell and then transfected one hour later with 6 µg of plasmid pHA-GPN using Lipofectamine-2000 (Invitrogen). At 72 h post-infection (h.p.i.), the cells were harvested, lysed by freeze-thaw cycling, sonicated, and used for recombinant virus screening using X-Gluc (5-bromo-4-chloro-3-indolyl-β-d-glucuronic acid; 800 μg/mL) as substrate for the product of the β-glucuronidase (β-GUS) marker gene. As described above for the MVA-gp145 and MVA-GPN viruses, during the purification steps, the β-GUS marker gene was deleted by homologous recombination.

The resulting MVA-based recombinant viruses were expanded in CEF cells and the crude preparations that were obtained were used for the propagation of the viruses in large cultures of CEF cells, followed by virus purification through two 36% (*w*/*v*) sucrose cushions and the virus titers were determined by the immunostaining plaque assay in DF-1 cells, as previously described [16]. The titer determinations of the different viruses were performed at least three times.

### 2.4. PCR Analysis of MVA-Based Recombinant Viruses

To test the identity and purity of the different MVA-based recombinant viruses, viral DNA was extracted from DF-1 cells that were infected at 5 pfu/cell with MVA-WT, MVA-gp145, MVA-GPN or MVA-gp145-GPN for 24 h. The cell membranes were disrupted by proteinase K treatment (0.2 mg/mL proteinase K in 50 mM Tris-HCl pH 8, 100 mM EDTA pH 8, 100 mM NaCl, 1% SDS; 1 h, 55 °C), followed by incubation with RNase A (80 µg/mL). Viral DNA was precipitated using 2-propanol. Primers TK-L: 5’-TGATTAGTTTGATGCGATTC-3’ and TK-R: 5’-CTGCCGTATCAAGGACA-3’ spanning TK flanking regions and primers HA-MVA: 5’-TGACACGATTACCAATAC-3’ and HA-II: 5’-GATCCGCATCATCGGTGG-3’ spanning HA flanking regions were used for PCR analysis of TK and HA loci, respectively. The amplification reactions were carried out with Phusion High-Fidelity DNA polymerase (BioLabs, Ipswich, MA, USA), according to the manufacturer´s recommendations.

### 2.5. Analysis of Virus Growth

To determine the growth profile of the different MVA-based recombinant viruses, monolayers of primary CEF cells that were grown in 12-well plates were infected at 0.1 pfu/cell with MVA-WT, MVA-gp145, MVA-GPN or MVA-gp145-GPN. Following virus adsorption for 60 min at 37 °C, the inoculum was removed and the infected cells were incubated with fresh DMEM containing 2% FCS at 37 °C in a 5% CO_2_ atmosphere. At different times post-infection (0, 24, 48 and 72 h), the cells were harvested by scraping, centrifuged for 5 min at 3000 rpm, supernatant removed, 0.1 mL of complete DMEM added to the cellular pellet, freeze-thawed three times, and briefly sonicated. Virus titers in cell lysates were determined by immunostaining plaque assay in DF-1 cells using rabbit polyclonal anti-VACV strain WR (1:1000; CNB), followed by anti-rabbit-horseradish peroxidase (HRP) (1:1000; SIGMA).

### 2.6. Time-Course Expression of HIV-1 Proteins gp145 and GPN by Western-Blot Analysis

Time-course expression of gp145 and GPN proteins from single and double recombinant viruses was performed by Western-blot. Monolayers of DF-1 or HeLa cells that were grown in 24-well plates were infected at 5 pfu/cell with MVA-WT, MVA-gp145, MVA-GPN or MVA-gp145-GPN viruses. At different times post-infection (0, 4, 8 and 24 h), infected cells were collected, and the cells extracts were fractionated by 10% SDS-PAGE and then analyzed by Western-blot using rabbit anti-gp120 antibody (1:3000; CNB) to evaluate the expression of gp145 protein, or rabbit anti-gag p24 antibody (1:1000; NIBSC) to determine the expression of GPN protein. Goat anti-rabbit-HRP (1:5000; SIGMA) was used as secondary antibody. The immunocomplexes were detected by an enhanced chemiluminescence system (ECL, GE Healthcare, Chicago, IL, USA).

### 2.7. Genetic Stability of MVA-Based Recombinant Viruses

The stability of HIV-1 antigens that were expressed by the different recombinant viruses was analyzed by serial passages in CEF cells grown in T25 tissue culture flasks. Monolayers of CEF cells were infected with MVA-gp145, MVA-GPN or MVA-gp145-GPN at 0.05 pfu/cell, and after 48–72 h the infected cells were collected by scrapping, freeze-thawed three times, and cellular extract was used to infect a monolayer of CEF cells for a new round of infection. This same procedure was repeated until seven serial passages were performed. Additionally, monolayers of DF-1 cells that were grown in 6-well culture plates were infected with serial dilutions of the virus stock from passage 7. After 1 h of viral adsorption, the inoculum was removed and the infected cells were overlayed with 1.9% Agar:DMEM 2X:2% FCS. When lysis plaques were visible, 20–25 isolated plaques were picked up. The correct expression of gp145 and GPN antigens both in passages P1 to P7 and in isolated plaques from P7 was analyzed by Western-blot using rabbit anti-gp120 or anti-gag p24 antibodies.

### 2.8. Fractionation of HIV-1 gp145 and Gag Antigens into Different Compartments after Detergent Treatment of Purified Recombinant MVA Particles

The localization of HIV-1 gp145 and Gag antigens in MVA recombinants was analyzed by sequential detergent treatment, as previously described [17]. Briefly, virions that were purified by sucrose gradients were resuspended by sonication in 0.1 mL of Tris-buffer (50 mM Tris-HCl pH 8.5, 10 mM MgCl_2_) containing the nonionic detergent NP-40 (1%). This and the following treatments were performed for 30 min at 37 °C. The E1 fraction (soluble lipid envelopes) was discarded by centrifugation, and the remaining pellet fraction was resuspended in 0.1 mL of Tris-buffer containing 1% NP-40 plus 50 mM DTT. The E2 fraction (soluble protein matrix-like membranes) was collected after centrifugation and the pellet was resuspended in 0.1 mL of the previous buffer with the addition of 0.5% DOC and 0.1% SDS. The soluble core proteins (E3 fraction) were removed by centrifugation, and the pellet fraction containing the remaining cores (C fraction) was resuspended in 0.1 mL of milliQ H_2_O. The collected fractions were resolved by SDS-PAGE under reducing conditions and the HIV-1 gp145 and Gag proteins were identified by Western-blot with specific antibodies.

### 2.9. Detection of Gag-Induced VLPs by Electron Microscopy Analysis

To detect HIV-1 Gag-induced VLPs in the supernatant of infected cells, 150 × 10^6^ HeLa cells were infected with MVA-GPN or MVA-gp145-GPN viruses at 5 pfu/cell for 24 hours. The VLPs were pelleted from infected cell culture supernatants by ultracentrifugation through a 20% sucrose cushion at 25,000 rpm in a Beckman SW28 rotor for 3 h. The VLP-containing pellet was resuspended in PBS buffer, loaded onto a 20%–60% *w*/*v* sucrose gradient, and then centrifuged at 35,000 rpm in a Beckmann SW41 rotor for 18 h. The gradient was fractionated from the top in 500 μL aliquots and then analysed by Western-blot to detect the presence of HIV-1 Gag specific p24 protein. The fraction of each virus that exhibited the highest Gag expression level was dialyzed to eliminate sucrose and salts through a 0.025 μm pore size membrane filter (Merck Millipore, Burlington, MA, USA) overnight at 4 °C. After dialysis, 100 μL of each sample was fixed with 4% paraformaldehyde for 30 min at 4 °C. For the negative staining of VLPs, 20 µL of each sample was adsorbed to carbon-coated collodion films that were mounted on 400-mesh/inch nickel grids (Aname, Madrid, Spain), floated two times in NH_4_Cl drops, washed several times with PBS, and stained with 2% uranyl acetate (Aname) for 1 min. For the immunogold labelling to detect gp145 that is associated to the VLP, the samples were incubated with a rabbit polyclonal anti-gp120 primary antibody (1:20; CNB), followed by an anti-IgG secondary antibody that was coupled to 10-nm colloidal gold beads (1:40; BBI Solutions, Crumlin, UK) and then stained with 2% uranyl acetate. Finally, VLP samples from negative staining or immunogold labelling were analysed in a transmission electron microscope (JEOL JEM-1011; CNB-CSIC Electron Microscopy Service, Madrid, Spain) that was equipped with an ES1000W Erlangshen charge-coupled-device (CCD) camera (Gatan Inc., Pleasanton, CA, USA) at an acceleration voltage of 40 to 100 kV.

### 2.10. Detection of HIV-1 Clade C ZM96 gp145 Protein on the Surface of Infected Cells

The presence of membrane-bound trimeric gp145 protein on the surface of non-permeabilized infected cells was assessed by flow cytometry using a panel of human broadly neutralizing antibodies (bNAbs) targeting quaternary V1/V2 N-glycans (PG9, PGT145 and PG16), V3 N-glycans (10-1074 and PGT121), outer domain (OD)-glycans (2G12), CD4 binding site (VRC01, VRC03 and b12), or gp120/gp41_ECTO_ interface (PGT151 and 35O22) epitopes on the native Env protein. bNAbs were obtained through the AIDS Reagent Program, Division of AIDS, NIAID. HeLa cells were infected with MVA-gp145, MVA-gp145-GPN or MVA-GPN recombinant viruses at 3 pfu/cell. At 16 h.p.i., the cells were rinsed with PBS (no calcium/magnesium), dissociated with 2 mM EDTA in 1X PBS, washed with FACS buffer (1% BSA, 2 mM EDTA in 1X PBS), and pelleted at 1500 rpm for 5 min. Cells were then stained with live/death fixable red dye (1:200, Invitrogen) for 30 min at 4 °C in the dark, washed twice with FACS buffer, and then blocked with 3% BSA for 30 min at 4 °C. 10 µg/mL in 50 µL FACS buffer of each primary human IgG anti-Env bNAb were used to stain 10^6^ cells for 30 min at 4 °C in the dark. The cells were then washed twice with FACS buffer and secondary F(ab’)2-goat anti-human IgG (H + L)-PE antibody (1:200, Beckman Coulter, Brea, CA, USA) in 50 µL FACS buffer was added onto the cells. After 30 min incubation at 4 °C in the dark, the cells were washed twice with FACS buffer and fixed with 0.5% formaldehyde. The samples were acquired in a GALLIOS flow cytometer (Beckman Coulter) and data analyses were performed using FlowJo software (Version 10.4.2; Tree Star, Ashland, OR, USA). Geometric Mean Fluorescence Intensity (gMFI) values on the “live cells” gate were used to analyse the results.

### 2.11. Peptides

The HIV-1 clade C ZM96 gp140 peptides were provided by the Centralised Facility for AIDS Reagents, NIBSC, UK. They spanned the HIV-1 gp140 from clade C (ZM96) that was included in the recombinant viruses MVA-gp145 and MVA-gp145-GPN as consecutive 15-mers overlapping by 11 amino acids. They were pooled in three different Env peptide pools: Env-1 (61 peptides), Env-2 (65 peptides), and Env-3 (40 peptides). The HIV-1 clade C ZM96 Gag peptide pool was provided by the NIH AIDS Reagent Program (Maryland, USA) and it spanned the HIV-1 Gag region from clade C (ZM96) included in MVA-GPN and MVA-gp145-GPN viruses as consecutive 20-mers overlapping by 10 amino acids. The HIV-1 clade C CN54 Pol and Nef peptide pools were provided by the EuroVacc Foundation (Lausanne, Switzerland) and spanned the HIV-1 Pol and Nef antigens from clade C (CN54) included in MVA-GPN and MVA-gp145-GPN viruses as consecutive 15-mers overlapping by 11 amino acids. For the analysis of the HIV-1 Pol-Nef-specific cellular immune responses, the following peptide pools were used: Gag-Pol, Pol-1, Pol-2 and Nef. The vaccinia virus (VACV) E3_140–148_ peptide (VGPSNSPTF; CNB-CSIC Proteomics Service), described as an immunodominant epitope in BALB/c mice [18], was used to detect the VACV-specific CD8 T cell responses.

### 2.12. Ethics Statement

The Ethical Committee of Animal Experimentation (CEEA) of Centro Nacional de Biotecnología (CNB-CSIC, Madrid, Spain) approved the animal experimental protocols, according to International EU Guidelines 2010/63/UE on protection of animals used for experimentation and other scientific purposes, Spanish National Royal Decree RD 1201/2005 and Spanish National Law 32/2007 on animal welfare, exploitation, transport and sacrifice (permit number PROEX 281/16).

### 2.13. Mouse Immunization Schedule

Groups of female BALB/c mice (6–8 week-old) purchased from ENVIGO (*n* = 5) were immunized with 1 × 10^7^ pfu of MVA-WT, MVA-gp145, MVA-GPN or MVA-gp145-GPN by bilateral intramuscular (i.m.) route. Three weeks later, the animals were immunized with MVA constructions as in the prime and 10 days after the last immunization, mice were sacrificed, and the spleens and draining lymph nodes (DLNs) were processed for Intracellular Cytokine Staining (ICS) assay and sera harvested for Enzyme-Linked Immunosorbent Assay (ELISA) to measure the cellular and humoral adaptive immune responses against HIV-1 or VACV antigens, respectively.

### 2.14. Analysis of the Cellular Immune Responses by Intracellular Cytokine Staining (ICS) Assay

#### 2.14.1. Analysis of HIV-1- and VACV-Specific CD4 and CD8 T Cell Responses

To determine the magnitude and phenotype of the HIV-1- or VACV-specific CD4 and CD8 T cell immune responses, 2 × 10^6^ splenocytes (erythrocyte-depleted) were seeded on 96-well plates and stimulated for 6 h in supplemented RPMI 1640 medium with 10% FCS, 1 µL/mL Golgiplug (BD Biosciences, Franklin Lakes, NJ, USA), anti-CD107a-FITC (BD Biosciences) and 5 µg/mL of HIV-1 clade C ZM96 gp140 (Env-1 + Env-2 + Env-3) or Gag peptide pools or 1 µg/mL of HIV-1 clade C CN54 Pol-Nef (Gag-Pol + Pol-1 + Pol-2 + Nef) peptide pools or 10 µg/mL of VACV E3 peptide. After stimulation, the splenocytes were washed, stained for surface markers, permeabilized (Cytofix/Cytoperm kit; BD Biosciences), and stained intracellularly with appropriate fluorochromes. For the analysis of HIV-1- and VACV-specific CD4 and CD8 T cell responses, the following fluorochrome-conjugated antibodies were used: CD3-PECF594, CD4-APCCy7, CD8-V500, CD107a-FITC, IL-2-APC, IFN-γ-PeCy7 and TNF-α-PE for functional analyses and CD127-PerCPCy5.5 and CD62L-Alexa700 for the phenotypic analyses. All of the antibodies were from BD Biosciences. The dead cells were excluded using the violet LIVE/DEAD stain kit (Invitrogen).

#### 2.14.2. Analysis of HIV-1-Specific Tfh Cell Responses

To analyze the magnitude and phenotype of the HIV-1-specific Tfh cell immune responses, 2 × 10^6^ splenocytes (erythrocyte-depleted) were seeded on 96-well plates and stimulated for 6 h in supplemented RPMI 1640 medium with 10% FCS, 1 µL/mL Golgiplug (BD Biosciences), anti-CD154 (CD40L)-Biotin (BD Biosciences) and 5 µg/mL of HIV-1 clade C ZM96 gp140 (Env-1 + Env-2 + Env-3) or Gag peptide pools or 1 µg/mL of HIV-1 clade C CN54 Pol-Nef (Gag-Pol + Pol-1 + Pol-2 + Nef) peptide pools. After stimulation, the splenocytes were washed, stained for surface markers, permeabilized (Cytofix/Cytoperm kit; BD Biosciences), and stained intracellularly with appropriate fluorochromes. For the analysis of HIV-1-specific Tfh cell responses, the following fluorochrome-conjugated antibodies were used: CD4-Alexa700, CD8-V500, CD154 (CD40L)-Biotin/Avidin-PE, IL-4-Alexa488, IFN-γ-PeCy7 and IL-21-APC for functional analyses and CXCR5-PECF594, PD1 (CD279)-APCefluor780 and CD44-PeCy5 (SPRD) for phenotypic analyses. All of the antibodies were from BD Biosciences. The dead cells were excluded using the violet LIVE/DEAD stain kit (Invitrogen).

#### 2.14.3. Analysis of gp140-Specific Germinal Center (GC) B Cell Responses

For the analysis of the magnitude and phenotype of the HIV-1 gp140-specific B cell immune responses, 2 × 10^6^ cells from the draining lymph nodes were seeded on 96-well plates, centrifuged, and the dead cells were stained by incubation with Fixable Viability Stain 520 (FVS 520) (BD Biosciences). After blocking the Fc receptors with anti-CD16/CD32 (BD Biosciences), the cells were incubated with 0.3 µg/10^6^ cells of biotinylated ZM96gp140 protein (University of Regensburg, Regensburg, Germany; Prof. Dr. Ralf Wagner) for 30 min. at 4 °C in the dark. After washing, the cells were stained with the following fluorochrome-conjugated antibodies for surface markers: CD3-FITC, B220-PECy7, IgD-APCH7, CD38-PerCPCy5.5, IgG1-BV421, GL7-Alexa647, IgM-PECF594 and CD19-Alexa700 (all from BD Biosciences).

For the analysis of the HIV-1- and VACV-specific T and B cell responses, cells from immunized animals were acquired in a GALLIOS flow cytometer (Beckman Coulter) and analyses of the data were performed using FlowJo software (Version 10.4.2; Tree Star, Ashland, OR, USA). Lymphocyte-gated events ranged between 10^5^ and 5 × 10^5^. After lymphocyte gating, Boolean combinations of single functional gates were generated using FlowJo to quantify the frequency of each response based on all the possible combinations of differentiation markers or cytokine expression. For each specific functional combination, background responses in the non-stimulated controls (RPMI) were subtracted from those that were obtained in stimulated samples, and the percentages of cytokine-producing cells in the control groups were also subtracted from the rest of the groups to eliminate the non-specific responses.

### 2.15. Antibody Measurement by Enzyme-Linked ImmunoSorbent Assay (ELISA)

As previously described, ELISA was used to assess antibody binding to gp140 and GPN proteins in serum [19]. Individual sera from immunized mice were two-fold serially diluted in duplicate and then incubated with 0.9 µg/mL of recombinant clade C ZM96gp140 purified protein (University of Regensburg, Regensburg, Germany; Prof. Dr. Ralf Wagner) or 1 μg/mL of the recombinant p17/p24 protein (C-clade consensus sequence; ARP695.1; NIBSC, Centralised Facility for AIDS Reagents, United Kingdom). The levels of binding anti-HIV-1 total IgG antibodies were defined as the last serum dilution that gave three times the mean OD_450_ value of the control group (end point titer). The IgG1, IgG2a or IgG3 antibody isotypes against Env were determined for each individual serum diluted at 1:64,000.

### 2.16. Data Analysis and Statistics

For the statistical analysis of flow cytometry data using a panel of human bNAbs, we report the geometric mean of the fluorescence and compute the 2.5% and 97.5% percentiles of the fluorescence distribution.

For the statistical analysis of the data obtained by ICS, an approach that corrects the values for the unstimulated controls (RPMI) and permits the calculation of confidence intervals and *p* values of hypothesis tests was applied [20]. Only antigen responses that were significantly higher than the corresponding RPMI values are depicted and the background that was obtained for the different cytokines in the non-stimulated controls never exceeded 0.05%. Analysis and presentation of the distribution of the polyfunctional responses were performed using SPICE version 5.1, which was downloaded from http://exon.niaid.nih.gov [21]. A comparison of distributions was performed using a Student’s T test and a partial permutation test, as previously described [21]. All of the values used for analyzing the proportionate representation of responses are background-subtracted.

For the statistical analysis of ELISA data, a one-way ANOVA test that was followed by Tukey’s honest significant difference criterion was performed.

## 3. Results

### 3.1. In Vitro Characterization of MVA-Based Recombinant Viruses Expressing HIV-1 Antigens

#### 3.1.1. Purity and Virus Growth

The correct insertion of HIV-1 antigens into MVA genome as well as the purity of the single and double recombinant viruses generated was checked by PCR using primers that cover the flanking regions of TK or HA loci. As shown in Figure 1A, the *env* and *gag-pol-nef* genes were successfully inserted into the viral TK or HA loci and no parental contamination was present in MVA-gp145, MVA-GPN and MVA-gp145-GPN virus stocks. These results were also confirmed by DNA sequence analysis.

Next, a growth curve analysis was carried out in permissive CEF cells to determine whether the insertion of HIV-1 genes affected the replication and virus growth of the different MVA-based recombinant viruses in cell culture. As observed in Figure 1B, the kinetics of viral growth were similar between the parental MVA-WT and the recombinant viruses MVA-gp145, MVA-GPN and MVA-gp145-GPN, confirming that the insertion of HIV-1 antigens into the MVA-WT genetic backbone did not impair its replication and growth properties.

#### 3.1.2. Time-Course Expression of HIV-1 Antigens and Genetic Stability of MVA-Based Recombinant Viruses

The expression kinetics of gp145 and GPN antigens were analyzed by Western-blot in cell extracts from DF-1 or HeLa cells that were infected with either the singles (MVA-gp145 and MVA-GPN) or the double MVA-gp145-GPN recombinant viruses. As shown in Figure 1C, the expected 145 kDa product (gp145) (upper panels) and the 55 kDa Gag product that were obtained by the processing of the Gag(ZM96)-Pol-Nef(CN54) polyprotein (lower panels) were detected over time, demonstrating that the generated MVA-based recombinant viruses correctly expressed the HIV-1 heterologous antigens.

The genetic stability of all the recombinants was assayed after seven serial passages of the viral stocks in CEF cells. Western-blot analysis of 20–25 individual virus plaques that were isolated from the last passage (passage 7) revealed that all of them (100%) correctly expressed the full-length gp145 and/or Gag antigens, thus confirming the stable integration of the transgenes into MVA genome and the correct expression of HIV-1 antigens after long-term passages (not shown, [22]).

#### 3.1.3. Incorporation of HIV-1 Antigens in Different Compartments within MVA Virions

Since poxvirus preparations that were used for vaccination purposes mostly consist of intracellular mature virus (IMVs), we analyzed whether virus particles that were isolated from MVA-infected cells incorporated HIV-1 Env and Gag antigens. To this aim, we determined the localization of both HIV-1 antigens within virion particles after the sequential detergent disruption of purified MVA preparations. As it is observed in Figure 1D, the Env protein is mainly found in the E1 (virus membrane) and C (insoluble core) fractions in both MVA-gp145 and MVA-gp145-GPN purified viruses, while Gag antigen is mainly observed in the insoluble C fraction in MVA-GPN or MVA-gp145-GPN preparations.

#### 3.1.4. Detection of Gag-Induced Virus-Like Particles (VLPs) by Electron Microscopy Analysis

HIV-1 Gag-induced VLPs were purified from the supernatants of HeLa cells that were infected with MVA-GPN or MVA-gp145-GPN and then identified by electron microscopy. As shown in Figure 2A–D, characteristic VLPs structures with a diameter of 150 nm were detected in both supernatants. Furthermore, the presence of Env on the surface of the VLPs from MVA-gp145-GPN supernatant was analyzed by immune electron microscopy using a polyclonal antibody to HIV-1 Env, followed by an anti-IgG secondary antibody coupled to colloidal gold beads. As it is observed in Figure 2C, no gp120-specific gold grains were associated with VLPs that were produced by MVA recombinant virus expressing Gag alone, while VLPs-associated gold grains were detected in the MVA-gp145-GPN sample (Figure 2D).

#### 3.1.5. bNAbs Binding Profile to Membrane-Bound gp145(ZM96) Protein

To determine whether the Env protein expressed by either the single MVA-gp145 or the double MVA-gp145-GPN recombinant viruses expose relevant epitopes, we further evaluated the affinity of a panel of bNAbs to the gp145 protein expressed on the plasma membrane of infected cells by flow cytometry. The selected panel of bNAbs targets most of the vulnerable HIV-1 Env trimer epitopes described. As shown in Figure 2E, all of the bNAbs assayed, except PGT151, specifically recognized the gp145 protein expressed by either the single MVA-gp145 or the double MVA-gp145-GPN viruses, although with different affinities. The lack of PGT151 recognition is consistent with the Env being uncleaved, since the cleavage site between gp120 and gp41 was mutated in the gp145 *env* gene. bNAbs targeting V3 N-glycans (10-1074 and PGT121), OD N-glycans (2G12) or CD4 binding site (VRC01, VRC03 and b12) displayed higher gMFIs values than bNAbs recognizing quaternary epitopes, either at the V2 appex (PG9, PGT145 and PGT16) or at the gp120-gp41 interface (35O22).

Overall, these results indicated that the gp145 protein, as expressed either by the single MVA-gp145 or the double MVA-gp145-GPN viruses, was inserted into the plasma membrane exposing some critical domains that are recognized by bNAbs.

### 3.2. Immune Response Induced by Single and Double MVA-Based Recombinant Viruses in Mice

To examine whether the co-expression of optimized HIV-1 gp145 and GPN proteins by the double recombinant virus MVA-gp145-GPN lead to an enhancement of the antigen-specific T and B cell immune responses when compared to the single expression by MVA-gp145 or MVA-GPN viruses, we performed in vivo studies in mice using the homologous MVA prime/MVA boost immunization approach. BALB/c mice (*n* = 5) received two doses of 1 × 10^7^ pfu by intramuscular route following the immunization schedule that is depicted in Figure 3A. The adaptive HIV-1-specific CD4, CD8 and Tfh T cell responses, as well as the antigen-specific B cell responses, were evaluated ten days after the last immunization.

#### 3.2.1. Characterization of the HIV-1-Specific CD4 and CD8 T Cell Immune Responses Elicited in Mice

Since cellular immune responses have a pivotal role in the HIV-1 control, we first evaluated the HIV-1-specific CD4 and CD8 T cell responses that were elicited by the single and double MVA-based recombinants. Splenocytes from immunized animals were non-stimulated (RPMI) or stimulated *ex vivo* for 6 h with the following pools of overlapping peptides covering the HIV-1 antigens included in the viruses assayed: gp140(ZM96) pool, Gag(ZM96) pool and Pol-Nef(CN54) pool. After stimulation, the cells were stained with specific antibodies to identify T cell lineage (CD3, CD4, and CD8), degranulation (CD107a) and effector cytokines to define responding cells (IFN-γ, IL-2 and TNF-α). The percentages of T cells with the CD4 or CD8 phenotype that produced IFN-γ and/or IL-2 and/or TNF-α and/or CD107a established the overall CD4^+^ or CD8^+^ T cell immune responses.

As shown in Figure 3B, the HIV-1-specific T cell responses were clearly polarized towards the CD4 compartment in all of the immunization groups. In animals that were immunized with the double recombinant MVA-gp145-GPN, the CD4^+^ T cell responses were mainly directed against the Env pool, with a magnitude that is significantly higher than the one elicited by the single MVA-gp145 virus (*p* < 0.001). However, the magnitude of the Gag-specific CD4 T cell response was significantly lower than the one that was obtained in MVA-GPN-immunized animals (*p* < 0.001). No differences in the Pol-Nef-specific CD4 T cell responses were detected between MVA-gp145-GPN- and MVA-GPN-immunized animals (Figure 3B, left panel). On the other hand, the HIV-1-specific CD8 T cell responses were significantly lower but similarly distributed against the three antigens, with magnitudes comparable to those that were obtained in animals that received the single MVA-gp145 or MVA-GPN viruses (Figure 3B, right panel). HIV-1-specific CD4 and CD8 T cell responses were undetectable in the control group that was immunized with two doses of MVA-WT.

The quality of a T cell response can be characterized by the profile of cytokine production and cytotoxic potential. Therefore, the polyfunctional profile of the overall HIV-1-specific CD4 T cell responses was determined based on the analysis of IFN-γ, IL-2 and TNF-α secretion and surface mobilization of CD107a on activated T cells as an indirect marker of cytotoxicity. Thirteen different HIV-1-specific CD4 T cell populations were induced after immunization with the different viruses (Figure 3C). In all of the groups, the overall antigen-specific response was highly polyfunctional, with more than 70% of CD4^+^ T cells secreting two, three or four cytokines. CD4^+^ T cells producing CD107a + IFN-γ + IL-2 + TNF-α and IFN-γ + TNF-α were the most representative populations induced.

Additionally, we also examined the phenotype of the HIV-1-specific CD4 T cell responses by measuring the expression of CD127 and CD62L surface markers, which allows for the definition of the different T cell memory subpopulations: T central memory (TCM; CD127^+^CD62L^+^), T effector memory (TEM; CD127^+^CD62L^−^) and T effector (TE; CD127^−^CD62L^−^). As shown in Figure 3D, the overall HIV-1-specific CD4 T cell responses were mostly of TEM phenotype in all immunization groups.

Overall, these results revealed that the expression of optimized HIV-1 gp145 and GPN proteins by single or double MVA-based recombinants mainly induced highly polyfunctional HIV-1-specific CD4 T cell immune responses with the TEM phenotype. The magnitude of the Env-specific CD4 T cell responses that are induced in MVA-gp145-GPN-immunized animals was greater than that obtained with the single MVA-gp145 virus, while the Gag-specific CD4 T cell responses were lower than that obtained with the single MVA-GPN virus.

#### 3.2.2. Characterization of the HIV-1-Specific Tfh Cell Immune Responses Elicited in Mice

T follicular helper CD4 cells (Tfh) are essential for the development and maintenance of germinal center (GC) reactions, a critical process that promotes the generation of long-lived high affinity humoral immunity. The interaction between Tfh and B cells is mediated by cell-associated and soluble factors, including CD40L (CD154), ICOS, IL-21, IL-10 and IL-4 [23]. Since the frequency and quality of Tfh cells have been previously correlated with the development of broadly neutralizing antibodies (bNAbs) [24,25], here we decided to characterize this specific cellular subset in the spleen of immunized mice. Splenocytes were non-stimulated (RPMI) or stimulated ex vivo for 6 h with gp140(ZM96), Gag(ZM96) or Pol-Nef(CN54) peptide pools. Flow cytometry gating strategy followed for the identification of the Tfh subset is illustrated in Figure 4A. Frequencies of total CD4 T cells with Tfh phenotype (CXCR5^+^PD1^+^) were higher in animals that received the singles MVA-gp145 or MVA-GPN or the double MVA-gp145-GPN viruses as compared to the control MVA-WT group (*p* < 0.001) (Figure 4B). Afterwards, we decided to evaluate the HIV-1-specific Tfh response by quantifying the CD4 T cells with Tfh phenotype that produced CD40L and/or IL-21 and/or IL-4 and/or IFN-γ. Since about 70% of Tfh CD4 T cells non-stimulated (RPMI) or stimulated with the different HIV-1 peptide pools were positive for IL-21, whereas in the non-Tfh CD4 population, only 2% of the cells were IL-21^+^, the HIV-1-specific Tfh response was established by the percentage of Tfh CD4 T cells that produced IFN-γ and/or IL-4 and/or CD40L after peptide pool stimulation in comparison with non-stimulated cells (RPMI). As shown in Fig. 4C, in MVA-gp145-GPN-immunized animals, the HIV-1-specific Tfh response was equally distributed between Env, Gag and Pol-Nef pools. The magnitudes of Env- and Gag-specific Tfh responses were similar to those that were obtained in animals that received the singles MVA-gp145 or MVA-GPN viruses, respectively (Figure 4C).

#### 3.2.3. Characterization of the Env-Specific GC B Cell Immune Responses and Memory B cells (MBCs) Elicited in Mice

Germinal centers (GCs) are secondary lymphoid structures within B cell follicles where the B cells go through affinity maturation and class-switch recombination to generate high-affinity antibodies [26,27,28,29]. In this study, we characterized the B cell populations that were induced by the different MVA-based recombinant viruses in draining lymph nodes (DLNs) at 10 days post-boost by flow cytometry. The gating strategy used for the identification of GC B cells (GL7^+^CD38^−^), memory B cells (MBCs) (GL7^−^CD38^+^) and class-switched MBCs (IgD^−^) is illustrated in Figure 5A. Moreover, we used biotinylated gp140(ZM96) protein to identify the frequency of Env-specific GC B cells.

As shown in Figure 5B and C, the intramuscular administration of two doses of MVA viruses induced important percentages of GC B cells, MBCs and switch MBCs in DLNs, although some of the differences were observed between groups. A slight but significant increase in the frequency of GC B cells in mice that received MVA-gp145 or MVA-gp145-GPN viruses was detected when compared to those animals that received MVA-GPN or MVA-WT viruses (*p* < 0.001) (Figure 5B, left panel). Similar differences were observed in the frequencies of GC B cells with IgG1^+^IgM^−^ phenotype (Figure 5B, right panel). In addition, animals that were immunized with the double MVA-gp145-GPN virus exhibited the highest frequencies of switch MBCs (IgD^−^GL7^−^CD38^+^) with IgG1^+^IgM^−^ phenotype as compared with those that were immunized with either the single recombinants or the control MVA-WT virus (Figure 5C). Finally, the analysis of the HIV-1 gp140-specific GC B cells revealed, as expected, that, in mice immunized with the single MVA-gp145 or the double MVA-gp145-GPN viruses, the percentages of Env-specific GC B cells were significantly higher (*p* < 0.001) than those that were detected in the groups that received MVA-GPN or MVA-WT viruses (Figure 5D, left panel). Notably, there were no differences in the levels of Env-specific GC B cells between the group immunized with the MVA-gp145-GPN virus expressing both gp145 and GPN antigens and the group immunized with MVA-gp145 that exclusively expressed the gp145 protein. We also evaluated the Gag-specific GC B cell responses using the biotinylated p17/p24 protein (clade C consensus), but the values obtained were not significantly different from the control groups. These might reflect the differential recognition and processing of Gag-induced VLPs by antigen presenting cells (APC) as compared with Env antigen. Representative flow cytometry profiles of vaccine-induced Env-specific GC B cells in the different immunization groups are shown in Figure 5D, right panel.

#### 3.2.4. Characterization of the HIV-1-Specific Humoral Immune Responses Elicited in Mice

The recombinant virus MVA-gp145-GPN was deliberately designed to co-express an optimized gp145 protein that was able to produce membrane-bound trimers similar to the native HIV-1 surface glycoproteins and, at the same time, a GPN polyprotein that is processed to produce Gag-induced VLPs, with the final aim of inducing high-affinity binding and potent antibody responses to HIV-1 antigens. Having this in consideration, we next decided to characterize the humoral responses against HIV-1 Env and Gag antigens elicited by the double MVA-gp145-GPN virus when compared to those that were induced by the single MVA-based recombinants.

The reactivity of serum from individual animals against the purified trimeric gp140 or p17/p24 antigens was quantified by ELISA. As shown in Figure 6A (left panel), mice that received the single MVA-gp145 or the double MVA-gp145-GPN recombinant viruses exhibited high titers of total IgG binding antibodies against gp140, with magnitudes above 10^5^. No Env-specific antibody responses were observed in the MVA group expressing a non-related antigen (MVA-GPN). Moreover, mice that were immunized with MVA-gp145-GPN recombinant elicited levels of total IgG binding antibodies against p17/p24 antigen with an end-point titer of approximately 10^3^ that did not differ from those that were obtained in the sera from mice immunized with the single MVA-GPN virus (Figure 6A, right panel). No Gag-specific antibody responses were detected in the group receiving an MVA expressing a non-related antigen (MVA-gp145).

Finally, we performed a more detailed analysis of the Env-specific humoral response by the determination of IgG2a/IgG1 and IgG3/IgG1 ratios induced by the single MVA-gp145 or double MVA-gp145-GPN viruses. As it is observed in Fig. 6B, the IgG2a/IgG1 ratio in both of the groups is higher than the IgG3/IgG1 ratio, indicating that the gp140-specific humoral response is mainly mediated by the IgG2a subclass. However, no differences in this IgG2a/IgG1 ratio are observed between both groups.

Overall, these results demonstrated that the expression of gp145 and GPN antigens by single or double MVA-based recombinants elicited HIV-1-specific humoral responses after a prime/boost immunization regimen, with noticeably high titers of gp140-specific binding antibodies.

#### 3.2.5. VACV-Specific CD8 T Cell Immune Response

We also evaluated the anti-vector T cell response that was induced by either the singles (MVA-gp145 and MVA-GPN) or the double MVA-gp145-GPN viruses using the VACV E3 peptide specific for the CD8 T cell subset. As shown in Figure 7A, immunization with MVA-WT induced significantly higher (*p* < 0.001) VACV-specific CD8 T cell responses when compared with those that were obtained in animals immunized with the recombinant MVA viruses. The immunization with MVA-gp145-GPN induced the lowest VACV-specific CD8 T cell responses, suggesting an inverse correlation between HIV-1- and VACV-specific T cell immune responses. Despite the differences in magnitude, similar polyfunctional profile and memory phenotype of anti-vector response were observed in all of the immunization groups. The VACV E3-specific CD8 T cell responses were highly polyfunctional (Figure 7B), with more than 80% of CD8^+^ T cells exhibiting three or four functions. CD8^+^ T cells producing simultaneously CD107a+IFN-γ+TNF-α or CD107a+IFN-γ+TNF-α+IL-2 were the most representative populations induced by the different viruses, and this response was mostly of the TE and TEM phenotypes (Figure 7C).

## 4. Discussion

Over three decades since the discovery of HIV virus as the AIDS causal agent, the development of an effective vaccine remains one of the major public health challenges of our society. Even though life expectancy of HIV-infected individuals has significantly improved due to the implementation of antiretroviral therapies, the development of a highly effective HIV/AIDS vaccine is essential in order to ultimately eradicate the HIV. Recombinant poxvirus vectors (including MVA, ALVAC or NYVAC viruses) have been extensively used in preclinical and clinical trials as vaccine candidates against HIV-1, being demonstrated to be safe and immunogenic. However, an ideal immunogen that is capable of inducing high protection by immunization has yet to be identified.

In the present study, we described the generation and preclinical evaluation in mice of new single and double MVA-based vaccine candidates, which expressed optimized HIV-1 clade C membrane-bound gp145(ZM96) trimeric protein and/or Gag(ZM96)-Pol-Nef(CN54) (GPN) polyprotein processed to form Gag-induced virus-like particles (VLPs). The HIV-1 clade C is highly prevalent in developing countries, like India and Southern Africa, where eradication efforts are further complicated by economic and cultural challenges.

Since vaccine-induced protective immunity is critically determined by HIV-1 Env conformation, the gp145 antigen was designed to be expressed as a membrane-bound trimeric protein resembling the native conformation. It has been shown that soluble recombinant native-like Env trimeric proteins are able to induce strong HIV-1-specific B cell and antibody responses, enhancing the development of broadly neutralizing antibodies (bNAbs) [30]. Additionally, Gag, Pol and Nef antigens were designed to be expressed as Gag-induced VLPs with the aim of favouring both the B and T cell immune responses, as it has been previously described by Böckl and colleagues [31]. The inclusion of both HIV-1 antigens in the same vector aims to potentiate and broaden the HIV-1-specific immune responses and to avoid the use of two different recombinant viruses expressing each protein individually, which represents a significant advantage regarding production cost.

All of the recombinant viruses were highly stable in cell culture and expressed high levels of HIV-1 gp145 and/or Gag as VLPs. Furthermore, we demonstrated that the insertion of foreign genes into the viral genome did not affect the virus replication of the singles MVA-gp145 and MVA-GPN or the double MVA-gp145-GPN viruses, since they grew as efficiently as the parental MVA-WT in CEF cells. In addition, the VLPs contain Env bound to the particles, as visualized by immunogold electron microscopy of purified VLPs from MVA-gp145-GPN-infected cells. Moreover, gp145 and Gag antigens were also detected in purified MVA virions and they were localized in different compartments, with gp145 being mainly found in the virus-membrane fraction and Gag in the insoluble Core fraction. The HIV-1 antigens that were incorporated within the recombinant MVA virus particles is an added advantage, as they could also contribute to promote immune responses to these antigens soon after virus infection.

The presence of properly processed and folded Env on the cell surface of the infected cells was demonstrated by flow cytometry using a panel of bNAbs that bind to the relatively conserved and functionally relevant regions of Env. These epitopes, called vulnerability sites, were exposed on the membrane-bound gp145 protein that was expressed by the single MVA-gp145 or the double MVA-gp145-GPN viruses, as indicated by specific binding of almost all of the bNAbs assayed. A membrane-bound conformation of gp145 at the cell surface would likely induce more bNAbs than a cell-released soluble trimeric gp140 form.

In the BALB/c mouse model, we observed that MVA-gp145-GPN was highly immunogenic, inducing high and polyfunctional Env- and Gag-specific CD4 T cell immune responses. Similar results were obtained in an immunization study that was performed in rhesus macaques, where a mix of single recombinant NYVAC vectors expressing Env as a trimeric soluble protein or Gag-induced VLPs elicited large CD4^+^ T cell responses [10]. Interestingly, we observed that the co-expression of Env and GPN antigens by the double MVA-gp145-GPN virus enhanced the Env-specific CD4 T cell responses when compared with the single MVA-gp145 recombinant, which is possibly due to the additional incorporation of gp145 antigen into Gag-induced VLPs, but did not abrogate the specific responses against Gag, although the magnitude was significantly reduced as compared with the single MVA-GPN virus. Lower levels of HIV-1-specific CD8 T cell immune responses were detected in immunized animals, which is consistent with the inefficiency of protein antigens in engaging the class I cellular machinery that is needed to elicit effective CD8 responses. In the case of VACV-specific CD8 T cells, we observed a reduction of this type of cells in the double MVA recombinant vector as compared to MVA-WT and the single recombinant vectors, which should be advantageous in vaccination when booster doses with the same vector are needed.

In terms of humoral immune responses, MVA-gp145-GPN recombinant virus induced titers of anti-gp140 binding antibodies in the order of 10^5^. Remarkably, similar levels were recently described combining MVA and simian adenoviruses expressing native-like Env soluble trimers [32]. Another study in rabbits using a soluble recombinant HIV-1 Env protein trimer reported end point titers that were in the order of 10^4^–10^5^, depending of the immunization regimen used [33]. In both studies, animals developed NAbs against tier-1 viruses and, importantly, some immunization regimens also induced autologous tier-2 NAbs responses. Moreover, we have previously described that, in non-human primates, the combination of NYVAC vectors expressing cell-released gp140 and Gag-derived VLPs as a prime, with gp140 protein as a boost, induced high titers of Env-specific binding antibodies, HIV-1 neutralizing antibodies, and ADCC responses [10]. These studies demonstrated that immunization with trimeric Env glycoprotein could induce the generation of protective responses. Therefore, it is expected that the MVA vector co-expressing gp145 and Gag-derived VLPs would trigger similar protective responses to HIV-1 antigens, although further studies in rabbits or non-human primates should be conducted.

Follicular helper T cells (Tfh) were identified in year 2000 as the key helper CD4 T cell population that was responsible for providing help to B cells [24,34]. During the last years, this cellular subset has been extensively studied in the settings of vaccination or chronic HIV infection. HIV-1-specific circulating Tfh (cTfh), defined by IL-21 production, were induced at higher levels by the partly efficacious RV144 HIV vaccine than by other HIV-1 vaccine candidates that had shown no efficacy [35]. An association between the proportion of PD-1^+^ cTfh or PD-1^+^CXCR3^−^ cTfh, and the induction of broadly neutralizing antibodies was also reported in HIV-1-infected patients [36]. Furthermore, a high frequency of HIV-1-specific cTfh was recently associated with preserved memory B cell responses in HIV controllers [37].

Since all of these findings suggested a key role for the Tfh response in inducing protective responses against HIV, we decided to characterize, in detail, the frequencies of total and antigen-specific Tfh cells that were elicited by the double MVA-gp145-GPN vector in comparison with the singles MVA-gp145 and MVA-GPN viruses. We found that the homologous MVA/MVA immunization regimen was able to induce high numbers of CD4 T cells with Tfh phenotype (CXCR5^+^PD-1^+^), of which about 70% of Tfh cells stimulated or non-stimulated were positive for IL-21. Since this marker is critically involved in the Tfh cell help to GC B cells in their process of proliferation and mutation to increase affinity of B cell responses [24], this result suggested that MVA-based vectors might represent an advantageous platform to potentially activate this subtype of CD4 T cells. A recent study involving patients from a cohort of individuals who naturally control HIV-1 replication, reported that cTfh help likely contributes to the persistence of controller memory B cell responses, as the frequency of HIV-1-specific cTfh is correlated with the induction of HIV-1-specific antibodies in functional assays. Moreover, a positive correlation between Gag-specific cTfh and the induction of Env-specific IgG was observed [37]. MVA-gp145-GPN elicited HIV-1-specific Tfh responses with magnitudes that were similar to those that are induced by the singles MVA-gp145 or MVA-GPN viruses. These responses were primarily directed against Env pool, followed by Gag pool correlating with the HIV-1-specific humoral immunity elicited, since higher magnitudes of Env-specific than Gag-specific IgG antibodies were detected.

Since GC B cells are the target of Tfh help and also the main source of bNAbs, we analyzed this population using the different MVA-based vectors. High frequencies of total GC B cells were observed in the DLNs of animals that received parental or recombinant MVA viruses, and Env-specific GC responses were found in animals that were immunized with viruses expressing gp145. The simultaneous expression of gp145 and GPN by MVA-gp145-GPN virus did not affect the Env-specific GC responses, since similar levels were induced in MVA-gp145- or MVA-gp145-GPN-immunized animals. Our findings are promising, since it has been reported in a study with rhesus macaques that the frequency of Env-specific Tfh cells and GC B cells correlated with the subsequent development of NAbs after infecting the monkeys. Moreover, broader antibody neutralization was associated with greater affinity maturation in memory B cells, a process where Tfh cells are a key element [38].

The MVA vectors that are described here have added immune advantages over vectors that were derived from the NYVAC strain [9,11,13]. As previously described, both MVA and NYVAC vectors differ in the extent of activation of host innate immune signals, with higher induction of the cytokine/chemokine pattern and of other gene signatures by MVA over NYVAC [39,40]. In addition, the replication capacity of NYVAC in non-permissive human cells is more restricted than that of MVA [41]. Additionally, the MVA backbone has permitted, in a stable manner, the insertion of both HIV-1 antigens. Co-expression of a membrane-bound trimeric Env for the induction of bNAbs and Gag-induced VLPs to potentiate Gag-specific responses from MVA-gp145-GPN could play an active immune role in the control of viremia during acute or chronic HIV infection [42,43].

Overall, our study revealed that the MVA-gp145-GPN virus induced high and polyfunctional HIV-1-specific CD4 T cell responses and triggered antigen-specific Tfh and GC B cells that correlated with robust HIV-1-specific humoral responses. It represents a novel vector in the HIV vaccine field with significant immune features that are relevant for protection.

## Figures and Tables

**Figure 1 viruses-11-00160-f001:**
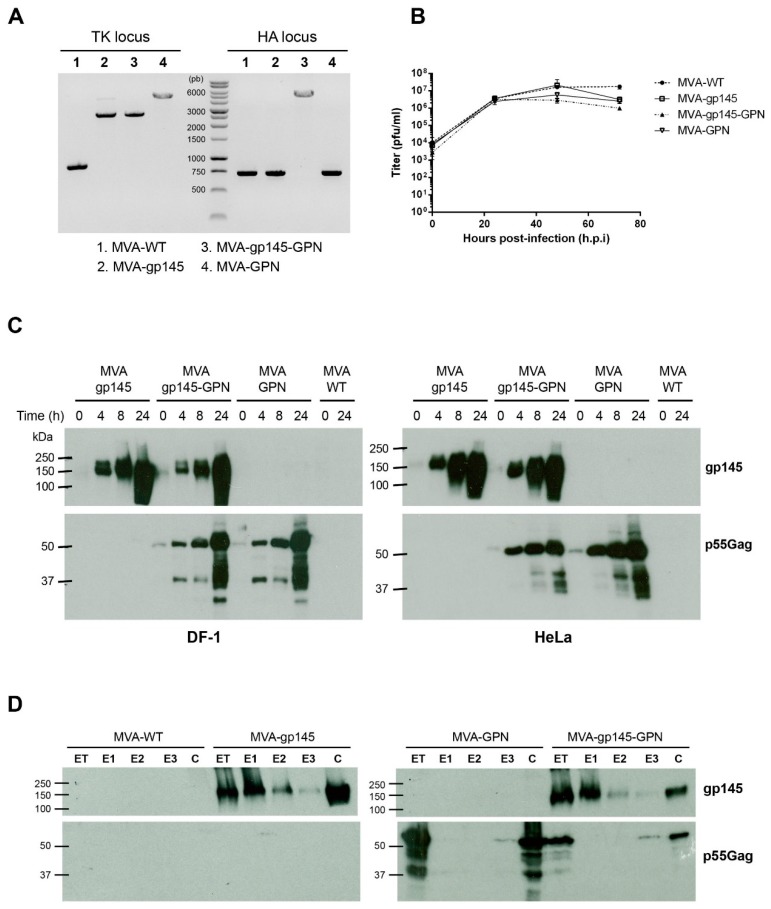
In vitro characterization of modified vaccinia virus Ankara (MVA)-based recombinant viruses expressing gp145 and/or Gag(ZM96)-Pol-Nef(CN54) (GPN) antigens. (**A**) Confirmation of HIV-1 gp145 and/or GPN insertion by PCR analysis. Viral DNA was extracted from DF-1 cells infected with MVA-WT, MVA-gp145, MVA-GPN or MVA-gp145-GPN at 5 pfu/cell. Primers thymidine kinase-L (TK-L) and TK-R spanning *J2R* (TK) flanking sequences or primers HA-MVA and HA-II spanning *A56R* (haemagglutinin (HA)) flanking sequences were used for PCR analysis of TK or HA loci, respectively. In parental MVA, a 853 bp-product corresponding to parental TK locus is obtained, while in MVA-gp145 and MVA-gp145-GPN a unique 2577 bp-product is observed and in MVA-GPN a unique band of 4500 bp is detected. Regarding the HA locus, a 750 bp-product is observed in MVA-WT, MVA-gp145 and MVA-GPN samples, while in MVA-gp145-GPN a unique 4500 bp-product is obtained. (**B**) Analysis of virus growth in a permissive cell line. Monolayers of CEF cells were infected with MVA-WT, MVA-gp145, MVA-GPN or MVA-gp145-GPN at 0.1 pfu/cell. At different times post-infection (0, 24, 48 and 72 h), the cells were collected and infectious viruses were quantified by immunostaining plaque assay in DF-1 cells. (**C**) Time-course expression of HIV-1 antigens by Western-blot analysis. Monolayers of DF-1 (left panels) or HeLa (right panels) cells were infected at 5 pfu/cell with MVA-WT, MVA-gp145, MVA-GPN or MVA-gp145-GPN viruses. At different times post-infection (0, 4, 8 and 24 h), the infected cells were collected, cells extracts fractionated by 10% SDS-PAGE, and analyzed by Western-blot using rabbit polyclonal anti-gp120 (upper panels) or anti-gag p24 (lower panels) antibodies to evaluate the expression of gp145 and Gag antigens, respectively. (**D**) Fractionation of viral proteins and localization of HIV-1 Env and Gag proteins within purified MVA-based recombinant viruses. Sucrose-purified virions from MVA-based recombinants were sequentially disrupted by treatment with detergents and different fractions were isolated, as described under Materials and Methods. The unfractionated lysate virions (total extract, ET) and the collected fractions (E1, E2, E3 and C) were analysed by Western-blot using anti-gp120 or anti-Gag antibodies.

**Figure 2 viruses-11-00160-f002:**
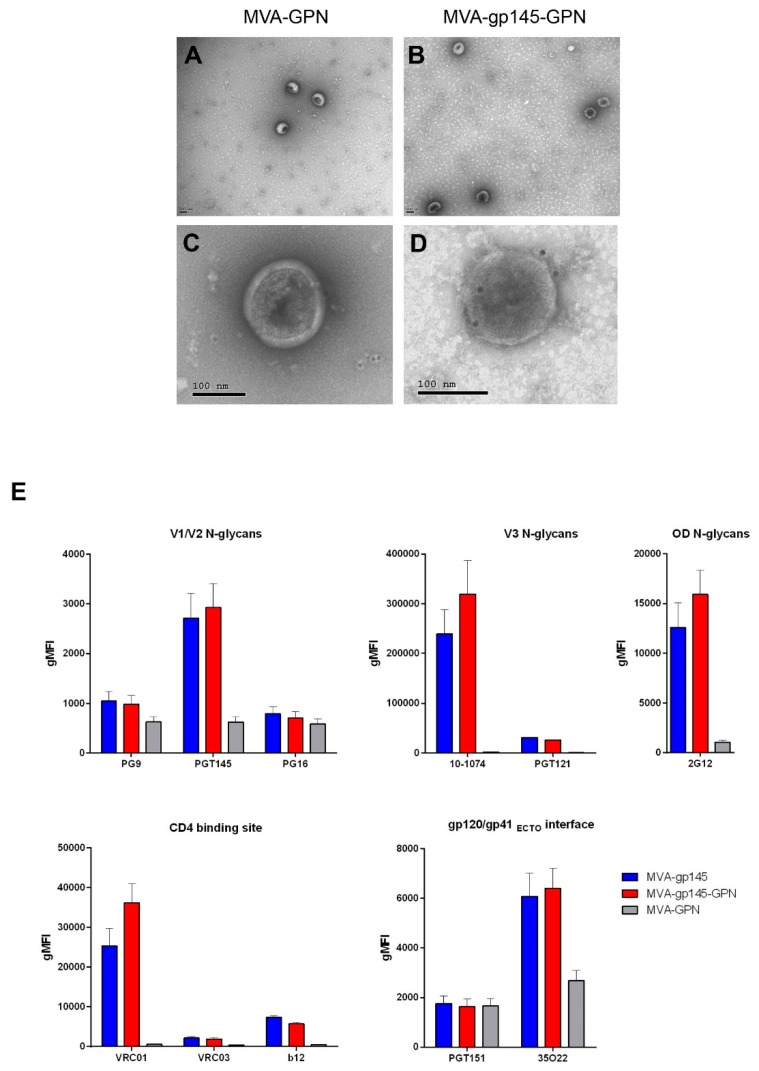
(**A**–**D**) Detection of Gag-induced virus-like particles (VLPs) by electron microscopy. HeLa cells were infected with MVA-GPN (**A**,**C**) or MVA-gp145-GPN (**B**,**D**) recombinant viruses at 5 pfu/cell and at 24 h.p.i. the supernatants were harvested, clarified, purified by ultracentrifugation through a 20% sucrose cushion, followed by a 20%–60% sucrose gradient, and then processed for negative staining (**A**,**B**) or immnogold labelling using anti-gp120 antibody (**C**,**D**), as described in Materials and Methods. Bar: 100 nm. (**E**) bNAbs binding profile to membrane-bound gp145 trimeric protein by flow cytometry. HeLa cells infected with MVA-gp145, MVA-gp145-GPN or MVA-GPN viruses were processed for flow cytometry, as described under Materials and Methods using 10 µg/mL of each primary human IgG anti-Env bNAb. The selected panel of human bNAbs targets quaternary V1/V2 N-glycans (PG9, PGT145 and PG16), V3 N-glycans (10–1074 and PGT121), outer domain (OD)-glycans (2G12), CD4 binding site (VRC01, VRC03 and b12) or gp120/gp41_ECTO_ interface (PGT151 and 35O22) on the native Env protein. Samples were acquired in a flow cytometer and geometric Mean Fluorescence Intensity (gMFI) values on the “live cells” gate were used to analyse the results. All of the bNAbs assayed, except PGT151, specifically recognized the gp145 protein expressed by either the single MVA-gp145 or the double MVA-gp145-GPN viruses, although with different affinities. bNAbs targeting V3 N-glycans, OD N-glycans or CD4 binding site exhibited higher gMFIs values than bNAbs recognizing quaternary epitopes either at V2 appex or at gp120-gp41 interface.

**Figure 3 viruses-11-00160-f003:**
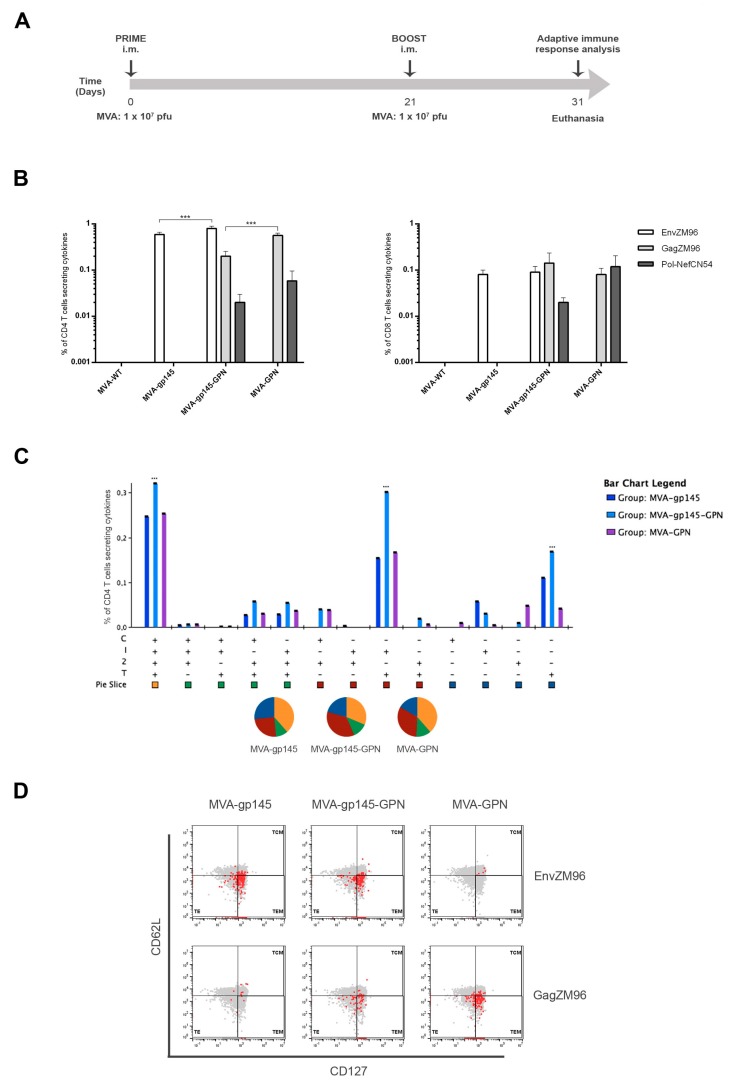
Polyfunctional HIV-1-specific CD4 and CD8 T cell immune responses elicited in spleen after prime/boost homologous immunization of mice with different MVA-based recombinant viruses expressing gp145 and/or GPN antigens. (**A**) Immunization schedule. Groups of 6–8-week-old female mice (*n* = 5) received the indicated doses of MVA-based recombinant viruses by bilateral intramuscular route (i.m.); three weeks later, the animals were immunized with MVA constructions as in the prime and 10 days post-boost, mice were sacrificed, and the spleens and draining lymph nodes (DLNs) were processed for Intracellular Cytokine Staining (ICS) assay and sera harvested for Enzyme-Linked Immunosorbent Assay (ELISA) to measure the cellular and humoral adaptive immune responses against HIV-1 or VACV antigens, respectively. (**B**) Magnitude of the HIV-1-specific CD4 (left) or CD8 (right) T cells at 10 days post-boost by ICS assay after the stimulation of splenocytes with the different HIV-1 clade C peptide pools. The total value of each group represents the sum of the percentages of HIV-1-specific CD4 or CD8 T cells secreting CD107a and/or IFN-γ and/or IL-2 and/or TNF-α against different HIV-1 peptide pools. Data are background-subtracted. ***, *p* < 0.001. (**C**) Polyfunctional profile of the overall CD4^+^ T cell response in the different immunization groups. The thirteen positive combinations of the responses are indicated on the *x* axis, while the percentages of the functionally different cell populations within the total CD4 T cells are represented on the *y* axis. Non-specific responses that were obtained in the control group were subtracted in all populations. Specific responses are grouped and colour-coded based on the number of functions. C: CD107a; I: IFN-γ; 2: IL-2; T: TNF-α. ***, *p* < 0.001. (**D**) Representative phenotypic profiles of the vaccine-induced memory CD4 T cell responses against Env(ZM96) or Gag(ZM96) peptide pools in the indicated immunization groups. The red dots indicate antigen-specific vaccine-induced CD4^+^ T cells overlaid on the total CD4 T cell subsets (grey).

**Figure 4 viruses-11-00160-f004:**
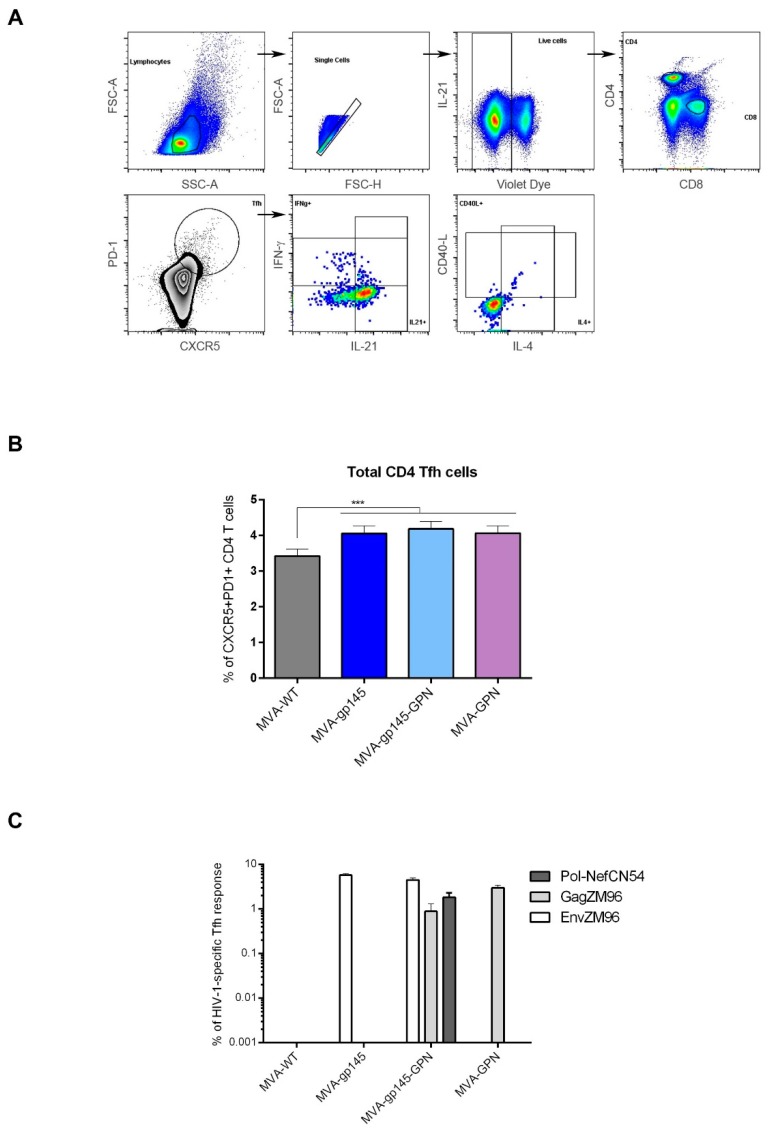
HIV-1-specific Tfh cell immune response elicited in spleen after prime/boost homologous immunization of mice with different MVA-based recombinant viruses expressing gp145 and/or GPN antigens. (**A**) Flow cytometry strategy for the identification of total (**B**) or HIV-1-specific (**C**) Tfh cells in splenocytes from immunized mice. Singlets were gated on lymphocytes followed by selection of live cells. CD4^+^CD8^−^ cells were then gated and analyzed based on the expression of the CXCR5 and PD-1 markers. The double positive CXCR5^+^PD-1^+^ population was used to identify total Tfh cells. HIV-1-specific Tfh cells were determined by the percentage of CXCR5^+^PD-1^+^ cells that produce IFN-γ and/or IL-4 and/or CD40L. FSC-A: forward-scatter area; FSC-H: forward-scatter height; SSC-A: side scatter area. (**B**) Magnitude of the total CD4 T cells with Tfh phenotype (CXCR5^+^PD1^+^) in spleen measured 10 days after the last immunization by ICS assay in non-stimulated (RPMI) lymphocytes derived from immunized animals. All the data are background-subtracted. ***, *p* < 0.001. (**C**) Magnitude of the HIV-1-specific Tfh cells in spleen. The total value in each group represents the sum of the percentages of Tfh^+^ T cells producing IFN-γ and/or IL-4 against HIV-1 peptide pools. All data are background-subtracted.

**Figure 5 viruses-11-00160-f005:**
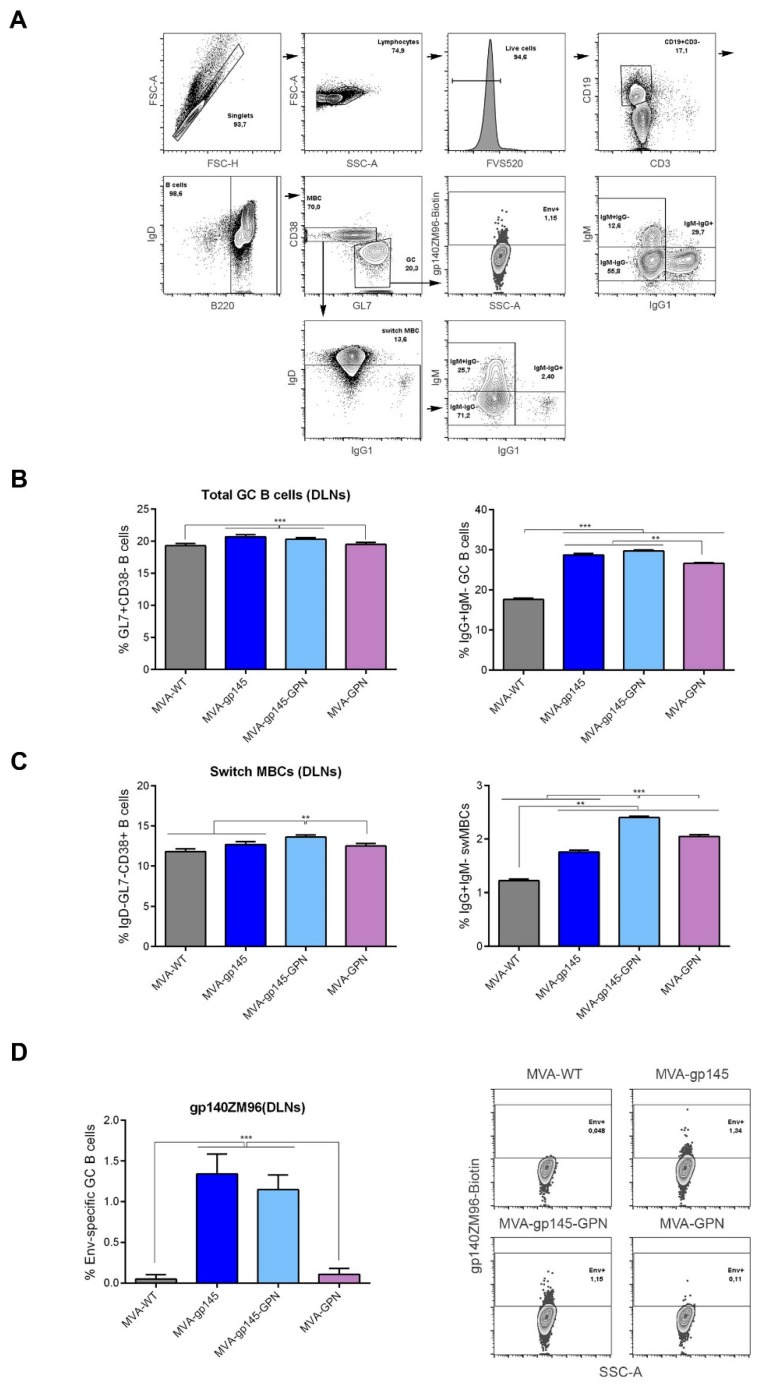
Env-specific B cell immune response elicited in DLNs after prime/boost homologous immunization of mice with different MVA-based recombinant viruses expressing gp145 and/or GPN antigens. (**A**) Flow cytometry strategy for the identification of total B cells (**B**), memory B cells (MBCs) (**C**) and Env-specific B cells (**D**) in lymphocytes from DLNs of immunized mice. B cells lymphocytes were identified as B220^+^ cells gated on singlets/live/CD3^−^CD19^+^ population. On B cells, we defined GC B cells (GL7^+^CD38^−^), memory B cells (MBCs) (GL7^−^CD38^+^) and class-switched MBCs (IgD^−^ and IgG1^+^IgM^−^) populations. Biotinylated gp140 protein was used to identify the frequency of Env-specific GC B cells. FSC-A: forward-scatter area; FSC-H: forward-scatter height; SSC-A: side scatter area. (**B**,**C**) Magnitude of the total GC B cells (**B**) or MBCs (**C**) in DLNs measured at 10 days post-boost by ICS assay in lymphocytes derived from immunized animals. **, *p* < 0.005; ***, *p* < 0.001. (**D**) Magnitude (left panel) and flow cytometry profiles (right panel) of the Env-specific GC B cells in DLNs measured at 10 days post-boost by ICS assay following the incubation of lymphocytes with biotinylated gp140(ZM96) protein. All data are background-subtracted. ***, *p* < 0.001.

**Figure 6 viruses-11-00160-f006:**
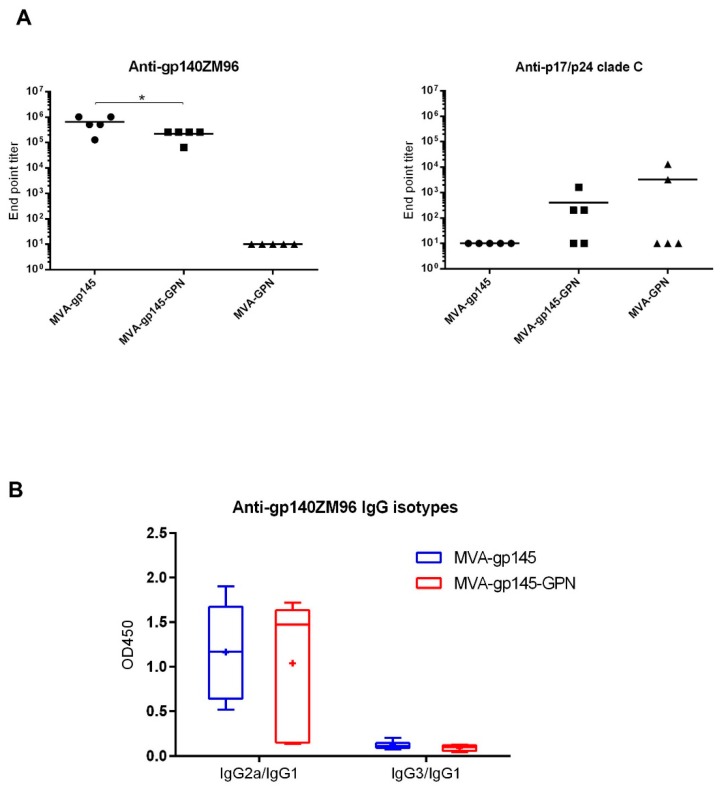
HIV-1-specific humoral immune response elicited in serum after prime/boost homologous immunization of mice with different MVA-based recombinant viruses expressing gp145 and/or GPN antigens. (**A**) Levels of anti-gp140- (left panel) or anti-p17/p24- (right panel) specific total IgG binding antibodies measured in individual sera from immunized mice at 10 days post-boost by ELISA. End point titer is defined as the last serum dilution that gave three times the mean OD_450_ value of the control group. The solid line represents the mean value of each group. *, *p* < 0.05. (**B**) Anti-gp140 IgG2a/IgG1 or IgG3/IgG1 ratios elicited in serum from immunized individual mice at 10 days post-boost measured as OD_450_ at a serum dilution of 1:64000 by ELISA. The “+” symbol indicates the mean value of each group; the box shows the 5th to 95th percentiles and the line in the box indicates the median value of each group.

**Figure 7 viruses-11-00160-f007:**
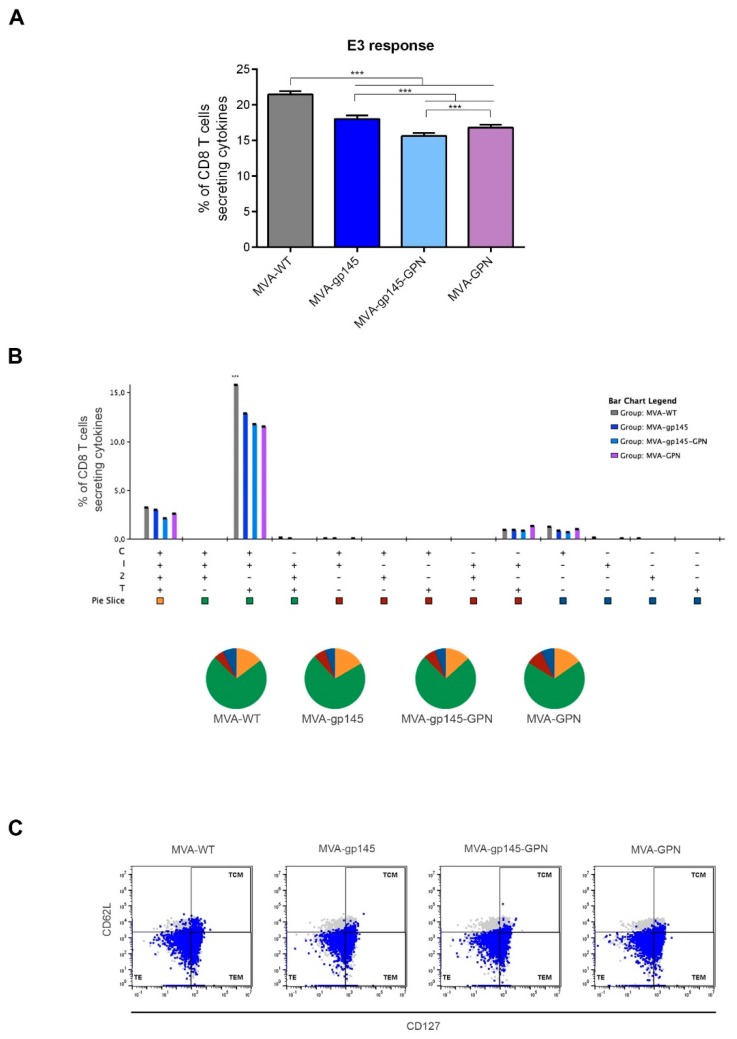
Polyfunctional VACV E3-specific CD8 T cell immune response elicited in the spleen after prime/boost homologous immunization of mice with different MVA-based recombinant viruses expressing gp145 and/or GPN antigens. (**A**) Magnitude of the VACV E3-specific CD8 T cells at 10 days post-boost by ICS assay after the stimulation of splenocytes with the VACV E3 peptide. The total value of each group represents the sum of the percentages of HIV-1-specific CD8 T cells secreting CD107a and/or IFN-γ and/or IL-2 and/or TNF-α against VACV E3 peptide. Data are background-subtracted. ***, *p* < 0.001. (**B**) Polyfunctional profile of the E3-specific CD8 T cell response in the different immunization groups. Distinct response combinations are indicated on the *x* axis, while the percentages of the functionally different cell populations within the total CD8 T cells are represented on the *y* axis. Specific responses are grouped and colour-coded based on the number of functions. C: CD107a; I: IFN-α; 2: IL-2; T: TNF-α. ***, *p* < 0.001. (**C**) Flow cytometry profiles of the vaccine-induced memory CD8 T cell phenotype against VACV E3 peptide in the indicated immunization groups.

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
