# Peer review of "A Novel MVA-Based HIV Vaccine Candidate (MVA-gp145-GPN) Co-Expressing Clade C Membrane-Bound Trimeric gp145 Env and Gag-Induced Virus-Like Particles (VLPs) Triggered Broad and Multifunctional HIV-1-Specific T Cell and Antibody Responses"

_viruses, 2019, doi:10.3390/v11020160_

Round 1
Reviewer 1 Report
The paper by Perdiguero et al. described the design, development and testing of novel MVA-based vaccines against HIV-1 (more specifically clade C). They developed MVA vectors expressing either gp145 or Gag-Pol-Nef polyprotein individually or together within the same vector (MVA-gp145-GPN). The authors present data on all the preliminary work they performed to design and produce the vaccine vectors, including all the data confirming that the vectors they rescued are as they are supposed to be. They also demonstrate that the vectors expressing the GPN polyprotein is capable of producing Gag-induced VLPs that possess gp145, when formed from MVA-gp145-GPN. They then went on to characterize the immune response induced by each of the vaccine vectors in a mouse model. Perdiguero and colleagues demonstrate that their vector expressing both the gp145 and GPN antigens is capable of inducing high and polyfunctional HIV-1-specific CD4 T cell responses and triggering antigen-specific Tfh and GC B cells that correlate with robust HIV-1-specific humoral responses.
This is a well written and very through paper. While the work is not highly innovative in that other poxviruses have been used as HIV-1 vaccine vectors expressing these same clade C antigens, the authors did demonstrate that they were able to produce a vector capable of expressing both antigens, which is an advantage when it comes to vaccine production costs.
Minor comments:
1. Can the authors please comment on why there is no Tfh response to the PolNef peptide pool in the MVA-GPN vaccinated animals in Figure 4C.
2. The reviewer suggests the authors reword lines 580-584 to indicate that it is expected that the gp140-specific GC B cell response is significantly different in gp145 expressing vectors compared to non-gp145 expressing vectors. This is to be expected. Also, can the authors comment on how/why there is any Env-specific GC B cell response in the MVA-GPN samples.
3. Please comment on why Gag-specific GC B cell responses were not performed for the readers knowledge.
4. The authors should point out that there were non-responders in the MVA-GPN and MVA-gp145-GPN vaccinated mice in regards to antibodies against p17/p24 (Figure 6). What does this mean in regards to the efficacy of your vaccine?
5. Line 717 – please change “while” to “but”.
Author Response
Reviewer 1
The paper by Perdiguero et al. described the design, development and testing of novel MVA-based vaccines against HIV-1 (more specifically clade C). They developed MVA vectors expressing either gp145 or Gag-Pol-Nef polyprotein individually or together within the same vector (MVA-gp145-GPN). The authors present data on all the preliminary work they performed to design and produce the vaccine vectors, including all the data confirming that the vectors they rescued are as they are supposed to be. They also demonstrate that the vectors expressing the GPN polyprotein is capable of producing Gag-induced VLPs that possess gp145, when formed from MVA-gp145-GPN. They then went on to characterize the immune response induced by each of the vaccine vectors in a mouse model. Perdiguero and colleagues demonstrate that their vector expressing both the gp145 and GPN antigens is capable of inducing high and polyfunctional HIV-1-specific CD4 T cell responses and triggering antigen-specific Tfh and GC B cells that correlate with robust HIV-1-specific humoral responses.
This is a well written and very through paper. While the work is not highly innovative in that other poxviruses have been used as HIV-1 vaccine vectors expressing these same clade C antigens, the authors did demonstrate that they were able to produce a vector capable of expressing both antigens, which is an advantage when it comes to vaccine production costs.
Minor comments:
1. Can the authors please comment on why there is no Tfh response to the PolNef peptide pool in the MVA-GPN vaccinated animals in Figure 4C.
Answer: The levels of PolNef-specific Tfh responses were not detected in the MVA-GPN group. Probably when GPN is co-expressed with gp145 the immunological environment and the secreted cytokines/chemokines favoured the detection of the PolNef-specific Tfh responses in MVA-gp145-GPN group while is undetected when GPN is expressed alone.
2. The reviewer suggests the authors reword lines 580-584 to indicate that it is expected that the gp140-specific GC B cell response is significantly different in gp145 expressing vectors compared to non-gp145 expressing vectors. This is to be expected.
Answer: As the reviewer suggested we reworded lines 580-584.
“Finally, the analysis of the HIV-1 gp140-specific GC B cells revealed, as expected, that in mice immunized with the single MVA-gp145 or the double MVA-gp145-GPN viruses, the percentages of Env-specific GC B cells were significantly higher (p<0.001) than those detected in the groups that received MVA-GPN or MVA-WT viruses (Fig. 5D, left panel)” Lines 610-614.
Also, can the authors comment on how/why there is any Env-specific GC B cell response in the MVA-GPN samples.
Answer: The Env-specific GC B cell response levels observed in MVA-WT and MVA-GPN are non-specific background values.
3. Please comment on why Gag-specific GC B cell responses were not performed for the readers knowledge.
Answer: We evaluated the Gag-specific GC B cell responses using biotinylated p17/p24 protein (clade C consensus) but the values obtained were not significantly different from the control groups (data not shown). These might reflect the differential recognition and processing of Gag-induced VLPs by antigen presenting cells (APC) compared with Env antigen. This paragraph was included in the manuscript (Lines 617-620 ).
4. The authors should point out that there were non-responders in the MVA-GPN and MVA-gp145-GPN vaccinated mice in regards to antibodies against p17/p24 (Figure 6). What does this mean in regards to the efficacy of your vaccine?
Answer: The levels of antibodies against p17/p24 were low in both MVA-GPN and MVA-gp145-GPN groups. In contrast to Env, the HIV-1 Gag protein is not exposed on the VLP membranes, therefore it is expected that antibody levels against Gag should be lower than to Env. Similar low antibody levels against Gag were previously described for NYVAC-GPN vector (ref 11). However, this differential response does not affect the efficacy of the vaccine since significant Gag-specific T cell responses (CD4 and CD8) were detected in both groups.
5. Line 717 – please change “while” to “but”.
Answer: As recommended we change “while” to “but”. (line 754)
Reviewer 2 Report
Perdiguero et al.
The authors describe the construction and testing of MVA vectors (more effective than previous NYVAC) that express trimeric Env, an improvement over gp120 expressed previously and GagPolNef with changes in Gag to improve processing to p55 which are referenced (although not fully explained), each expressed alone or together. The growth kinetics and protein expression by vectors is checked and sophisticated cellular, immunology flow cytometry analysis of immune responses in immunized balbc mice is reported, including a check of antigen specific Tfh and GC responses. It is interesting that anti-vector cellular immunity is inversely correlated with responses to the recombinant antigens. The discussion also puts the analysis into nice context with other findings.
Overall, the paper is well written and interesting. I do however, have a few comments below:
The authors appear to imply that Gag budding does not occur with unprocessed Gag, but in fact immature Gag can mediate virus release and later condense to form the p24 cone and other processed structures after release. So if I understand correctly, the modified Gag strategy correctly, it is unclear to me why improved processing would improve particle formation. This might simply require an additional sentence of explanation so that the reader is less obligated to read the cited reference to understand the concepts here.
The authors intend to elicit protective immunity depending on Env conformation, but there is not much coverage of neutralizing responses. It is not clear then what antibody responses are being sought (neutralizing or ADCC/ADCVI etc). There is an antigenic analysis showing "broad reactivity with bnAbs". However, this does not demonstrate native like conformation. There is no mention of reactivity with non-neutralizing antibodies that would really help to determine whether the trimers adopt a native conformation. Importantly, the lack of PGT151 recognition is consistent either with the Env being uncleaved (PGT151 is specific for cleaved trimer) and/or the lack of complex glycans by which the PGT151 antibody binds. It could also possibly indicate a simply lack of cross reactivity of PGT151 with the chosen strain. Tables of neutralization data are available online (e.g. CAPNAP) that help curate this information and would provide a way of correlating binding by the authors as shown in Fig 2 with neutralization IC50s.
Overall, the FACS data in Fig 2 don't really support a native conformation for surface Env, and certainly don't imply "favored exposure of bnAb epitopes" as no non-neutralizing antibodies included to verify the possibility of favored exposure. Section 1.5 and text elsewhere on this topic should be adjusted to acknowledge this point. Also, the data could be expanded to include a clade C reactive V3 antibody and a non-neutralizing CD4 binding site antibody like F105 or similar. It is possible that MVA expression is incompatible with native trimer expression, as gp120/gp41 processing appears absent (Fig 1). However, it is widely believed that gp120/gp41 processing is important for native trimer formation. The Env band in Fig 1 seems to be gp145 which would imply entirely uncleaved and also an additional larger band. Are these different glycan variants of uncleaved gp145? One solution might be to express trimers linked by a flexible linker between gp120 and gp41 to allow proper folding (e.g. NFL).
Gp145 is said to be "optimized" (e.g. line 604), but it is not clear what is actually optimized -does this mean it is codon optimized or is it mutated in some way aside from the truncation mutation?
"NCS" acronym of a method/product used for MVA infections should be defined
I am not completely clear on Fig 3C. What the red dots versus grey dots signify? - and if there is a control group MVA WT for this. Perhaps a sentence of additional explanation would help clarify.
Labels in Fig 5 and elsewhere in some figures are microscopic and should be enlarged for clarity.
Author Response
Reviewer 2
The authors describe the construction and testing of MVA vectors (more effective than previous NYVAC) that express trimeric Env, an improvement over gp120 expressed previously and GagPolNef with changes in Gag to improve processing to p55 which are referenced (although not fully explained), each expressed alone or together. The growth kinetics and protein expression by vectors is checked and sophisticated cellular, immunology flow cytometry analysis of immune responses in immunized balbc mice is reported, including a check of antigen specific Tfh and GC responses. It is interesting that anti-vector cellular immunity is inversely correlated with responses to the recombinant antigens. The discussion also puts the analysis into nice context with other findings.
Overall, the paper is well written and interesting. I do however, have a few comments below:
1. The authors appear to imply that Gag budding does not occur with unprocessed Gag, but in fact immature Gag can mediate virus release and later condense to form the p24 cone and other processed structures after release. So if I understand correctly, the modified Gag strategy correctly, it is unclear to me why improved processing would improve particle formation. This might simply require an additional sentence of explanation so that the reader is less obligated to read the cited reference to understand the concepts here.
Answer: As the reviewer suggested we included an additional sentence of explanation about the modified Gag strategy.
“ In the optimized HIV-1 GPN gene the natural ribosomal (−1) frameshift between Gag and Pol was restored to skew Gag:PolNef expression to approximately 10:1, and the N-terminal myristoylation signal was reintroduced to enable release of GagPolNef virus-like particles from infected cells (9)”. Lines 102-105.
2. The authors intend to elicit protective immunity depending on Env conformation, but there is not much coverage of neutralizing responses. It is not clear then what antibody responses are being sought (neutralizing or ADCC/ADCVI etc).
Answer: We have previously described that in-non human primates the combination of NYVAC vectors expressing cell-released gp140 and Gag-derived VLPs as a prime, with gp140 protein as a boost, induced high titers of Env-specific binding antibodies, HIV-1 neutralizing antibodies and ADCC responses (10). It is expected that the MVA vector co-expressing gp145 and Gag-derived VLPs would trigger similar protective immune responses to HIV-1 antigens although further experiments in rabbits or non-human primates will be needed. Lines 768-775.
3. There is an antigenic analysis showing "broad reactivity with bnAbs". However, this does not demonstrate native like conformation. There is no mention of reactivity with non-neutralizing antibodies that would really help to determine whether the trimers adopt a native conformation.
Answer: We highly appreciate the reviewer comments and experiments using non-neutralizing antibodies will be included in future studies. We have reworded those sentences that refer to gp145 adopt a native-like conformation indicating that the gp145 protein was inserted into the plasma membrane exposing epitopes that were recognized by broadly neutralizing antibodies (bNAbs). Lines 23, 244, 460, 471-472, 745 and 838.
Importantly, the lack of PGT151 recognition is consistent either with the Env being uncleaved (PGT151 is specific for cleaved trimer) and/or the lack of complex glycans by which the PGT151 antibody binds. It could also possibly indicate a simply lack of cross reactivity of PGT151 with the chosen strain. Tables of neutralization data are available online (e.g. CAPNAP) that help curate this information and would provide a way of correlating binding by the authors as shown in Fig 2 with neutralization IC50s.
Answer: The lack of PGT151 recognition is consistent with the Env being uncleaved, since the cleavage site between gp120 and gp41 was mutated. Lines 465-467.
Overall, the FACS data in Fig 2 don't really support a native conformation for surface Env, and certainly don't imply "favored exposure of bnAb epitopes" as no non-neutralizing antibodies included to verify the possibility of favored exposure. Section 1.5 and text elsewhere on this topic should be adjusted to acknowledge this point.
Answer: As we have not used non-neutralizing antibodies to verify the possibility of favored exposure the term "favored exposure of bNAb epitopes" has been changed in the manuscript as described above.
Also, the data could be expanded to include a clade C reactive V3 antibody and a non-neutralizing CD4 binding site antibody like F105 or similar. It is possible that MVA expression is incompatible with native trimer expression, as gp120/gp41 processing appears absent (Fig 1).
Answer: We highly appreciate the reviewer comments. MVA expression is not incompatible with native trimer expression. The gp120/gp41 processing appears absent because of the antigen design. Lines 465-467.
However, it is widely believed that gp120/gp41 processing is important for native trimer formation. The Env band in Fig 1 seems to be gp145 which would imply entirely uncleaved and also an additional larger band. Are these different glycan variants of uncleaved gp145? One solution might be to express trimers linked by a flexible linker between gp120 and gp41 to allow proper folding (e.g. NFL).
Answer: The additional larger bands are different glycan variants of uncleaved gp145
Gp145 is said to be "optimized" (e.g. line 604), but it is not clear what is actually optimized -does this mean it is codon optimized or is it mutated in some way aside from the truncation mutation?
Answer: Both. the gp145 sequence was codon-optimized and also the cleavage site between gp120 and gp41 was mutated to prevent cleavage but the details of the optimization product will be described elsewhere.
"NCS" acronym of a method/product used for MVA infections should be defined
Answer: NCS: Newborn Calf Serum. It was defined in line 92.
I am not completely clear on Fig 3C. What the red dots versus grey dots signify? - and if there is a control group MVA WT for this. Perhaps a sentence of additional explanation would help clarify.
Answer: The figure 3D represents the overlay of HIV-1 specific CD4 T cell responses (red dots) onto total CD4 T cells (grey dots). In this case, the single MVA-GPN recombinant is the control group for the Env-specific response whereas the single MVA-gp145 is the control group for Gag-specific response. As the reviewer suggested, the figure legend was modified for clarity.
“Figure 3D: Representative phenotypic profiles of the vaccine-induced memory CD4 T cell responses against Env(ZM96) or Gag(ZM96) peptide pools in the indicated immunization groups. The red dots indicate antigen-specific vaccine-induced CD4+ T cells overlaid on the total CD4 T cell subset (grey).” (lines 550-552)
Labels in Fig 5 and elsewhere in some figures are microscopic and should be enlarged for clarity.
Answer: As the reviewer suggested the labels were enlarged for clarity.